# Key drivers of cloud response to surface-active organics

S.J. Lowe[1,2], D.G. Partridge [3], J.F. Davies[4], K.R. Wilson[5], D. Topping[6] & I. Riipinen[1,2,7]*

Aerosol-cloud interactions constitute the largest source of uncertainty in global radiative forcing estimates, hampering our understanding of climate evolution. Recent empirical evidence suggests surface tension depression by organic aerosol to significantly influence the formation of cloud droplets, and hence cloud optical properties. In climate models, however, surface tension of water is generally assumed when predicting cloud droplet concentrations. Here we show that the sensitivity of cloud microphysics, optical properties and shortwave radiative effects to the surface phase are dictated by an interplay between the aerosol particle size distribution, composition, water availability and atmospheric dynamics. We demonstrate that accounting for the surface phase becomes essential in clean environments in which ultrafine particle sources are present. Through detailed sensitivity analysis, quantitative constraints on the key drivers – aerosol particle number concentrations, organic fraction and fixed updraft velocity – are derived for instances of significant cloud microphysical susceptibilities to the surface phase.

[1] Department of Environmental Science and Analytical Chemistry (ACES), Stockholm University, Stockholm, Sweden. [2] Bolin Centre for Climate Research, Stockholm University, Stockholm, Sweden. [3] College of Engineering, Mathematics and Physical Sciences, University of Exeter, Exeter, UK. [4] Department of Chemistry, University of California Riverside, Riverside, CA, USA. [5] Chemical Sciences Division, Lawrence Berkeley National Laboratory, Berkeley, CA, USA. [6] School of Earth, Atmospheric and Environmental Science, University of Manchester, Manchester M13 9PL, UK. [7] Aerosol Physics, Faculty of Science, Tampere University of Technology, Tampere, Finland. *email: ilona.riipinen@aces.su.se

Aerosol particles play a crucial role in the Earth's energy budget by both scattering and absorbing solar radiation (the direct effect)[1] and acting as cloud condensation nuclei (CCN) and ice-nucleating particles (INP) (the indirect effects)[2]. The complex organic fraction present in atmospheric aerosol is a major contributor to the uncertainty in aerosol–climate interactions, hindering our ability to make accurate climate predictions[3–5].

The thermodynamic viability of a given aerosol particle to act as a CCN is described by Köhler theory[6]. By simultaneously accounting for the curvature of the liquid–vapour interface (Kelvin effect) and the presence of solutes in the particle phase (Raoult effect), the CCN activity of an aerosol can be assessed at a given ambient water saturation ratio $S$ (or supersaturation $s = S - 1$). CCN activation occurs when the water vapour supersaturation $s$ exceeds the critical supersaturation $s_c$ at the droplet surface. Most earlier studies of ACI have been conducted assuming a surface tension of a pure water droplet and that solutes, including the organic compounds, reside in a single bulk aqueous phase[7,8]. It is widely acknowledged, however, that some atmospheric organic aerosol constituents form a distinct surface phase that leads to a reduction in the Raoult effect and surface tension, potentially lowering $s_c$[9–12].

Recent laboratory studies[13,14] indicate a previously undetected, but potentially significant, influence on CCN concentrations by surface-active organics. In particular, a compressed film representation, in which the organic molecules form a film on the growing cloud droplet surface, has been shown to reproduce observed CCN activity and droplet growth for a range of atmospherically relevant organic–inorganic mixtures[14]. Surface-active organic species have furthermore been identified in field samples of aerosol particles from various environments. These include, e.g., fatty acids present in sea spray[15], carboxylic acids present in secondary organic aerosol from forests[10,13] and organic compounds in nascent ultrafine mode (NUM) particles at a coastal location[16]. Specifically, significant CCN enhancements, up to tenfold as compared with the standard approaches, by surface tension suppression have been reported at Mace Head[16], Ireland, in spite of competing solute effects related to the surface composition. The broad applicability and climate relevance of these observations has not, however, been tested.

Non-linear responses of cloud formation to bulk-surface partitioning and surface tension depression complicate the understanding of ACI. It has been suggested that both effects need to be addressed simultaneously to avoid erroneous CCN and cloud droplet number concentration (CDNC) predictions[17]. In the studies to date, this has resulted in only minor changes in $s_c$, CCN or CDNC[12,18,19]. These studies, however, have not accounted for surface phenomena based on the new molecular models derived from recent experimental findings[13,16] or complex interdependencies between atmospheric dynamics, particle-size distribution shapes and chemical composition. Through consideration of the influence of surface tension depression relative to a standard Köhler formulation, across a range of fixed updraft velocities of 0–3.0 m s$^{-1}$, CDNC enhancements of 10–40% and 0–50% in marine and urban environments have been predicted in a cloud parcel model framework[20]. These results indicate that surface phenomena can play an important role under some conditions, but they remain to be verified in the light of recent findings using comprehensive sensitivity analyses.

Processes governing ACI need to be described with sufficient accuracy while maintaining computational feasibility in general circulation model (GCM) and Earth System Model (ESM) frameworks, with priority given to those with the largest impacts on cloud formation and properties. In GCMs and ESMs, the number concentration of activated aerosol particles, which is used in determination of cloud microphysical and optical properties, is parameterised in terms of the updraft velocity and aerosol-size distribution and composition[21,22]. Typically, droplet activation parameterisations contain drastic simplification of atmospheric aerosol composition due to computational limitations—organic aerosol species are routinely represented by a maximum of two tracers, for example[5]. Constraining uncertainties inherent to ACI requires new approaches to quantify the influence of organic compounds on cloud formation in various environments, characterised by the chemical composition[23] and size distribution[24] of aerosol particles. Cloud parcel models, while providing the theoretical basis for cloud droplet activation parameterisations used in climate models, also allow for detailed molecular-level descriptions of chemistry (including the most recent descriptions of surface phenomena arising from the recent observations[14,16]) and microphysics of aerosol particles and cloud droplets owing to their computational efficiency.

Here, we have embedded an approximation[16] of the compressed film model[14] as a novel description of surface-active organic molecules in a cloud parcel model to examine the applicability of recent findings in the larger context of climate, cloud formation and ACI (see Supplementary Fig. 1). We probe the complete atmospherically relevant parameter space representative of marine and boreal continental environments using the Sobol algorithm[25–27] for model variance decomposition, and demonstrate the multidimensionality of cloud responses to surface phenomena. With the cloud parcel model framework developed here, driven with observational data, we illustrate how the shape of the aerosol-size distribution, particle composition and updraft velocity constitute key drivers of surface phase induced cloud susceptibilities. We derive quantitative criteria for significant CDNC and cloud optical property responses to surface activity and identify environments, where detailed understanding of these phenomena is needed for accurately predicting ACI. Our work hence has the potential to facilitate the development of new and improved climate models that account for the molecular-scale phenomena where they matter most; and guide the experimental community to collect observational data in environments where detailed composition information is most needed.

## Results

**Cloud response to surface phase in three environments**. To assess the implications of the organic aerosol surface phase for cloud formation, and e.g., preindustrial radiative forcing estimates, we characterised marine (MA) and continental boreal (HYY) aerosol populations for simulation of cloud events with an adiabatic cloud parcel model. Each environment was represented by a submicron bimodal aerosol number concentration size distribution consisting of Aitken and accumulation modes, and mass fraction-based composition (see Fig. 1a, b and Table 1). In both environments, simple mixtures of inorganics and a single proxy organic compound were prescribed as approximations to real mixtures that likely contain hundreds, if not thousands, of different organic compounds[3]. In addition to the MA and HYY cases, which can be considered representative of large marine or continental boreal regions, further consideration is given to the NUM-event (NE) instance[16] to illustrate how strong CCN responses might manifest in explicitly simulated atmospheric conditions. For each case (MA, HYY and NE), an adiabatic cloud formation event was simulated twice, first with a single bulk phase Köhler theory description of cloud droplet growth (denoted BK), assuming complete dissolution of the organics, and second with an approximation[16] to the compressed film model[14], further denoted CF (see the Methods section, Fig. 1; Supplementary Fig. 1). While the BK approach represents a limiting case with

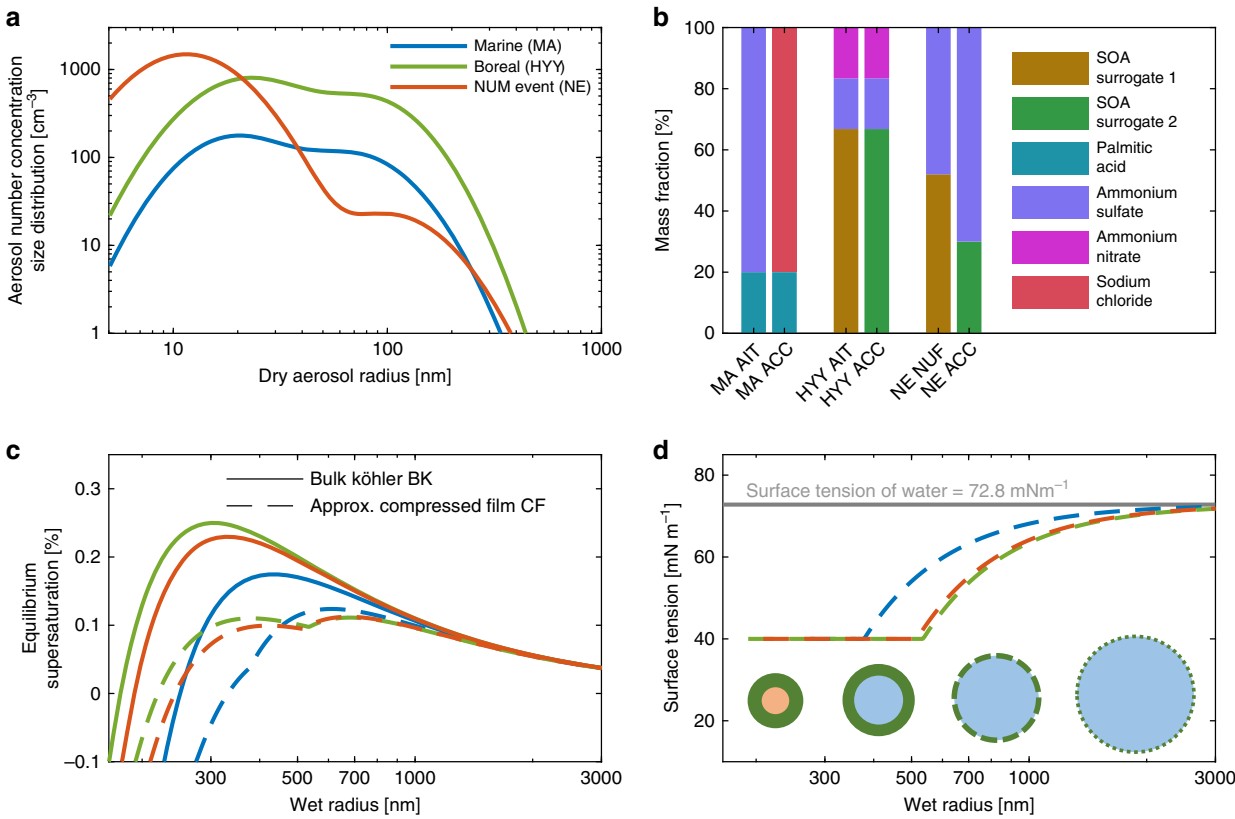

**Fig. 1** Characterisation of marine (MA), boreal (HYY) and nascent ultrafine mode event (NE) aerosol populations and their thermodynamic description by Bulk Köhler (BK) and compressed film (CF) models. **a** Dry aerosol number concentration size distributions for MA[52] (blue), HYY[31] (green) and NE[16] (orange) particle populations. **b** Assumed chemical composition of the particle populations. **c** Köhler curves corresponding to dry aerosol particles with 50 - nm radii and the compositions shown in **b**). Solid and dashed curves correspond to bulk Köhler (BK) and compressed film (CF) hydrometeor growth and phase structure descriptions, respectively. All Köhler curves are for Aitken mode compositions. **d** CF (dashed) and BK (solid, grey) surface tension curves corresponding to the 50 nm (in radius) Aitken mode particle Köhler curves displayed in **b**). Below the curves, a schematic illustrates the thinning and dissolution of the initially complete film-like surface phase due to hydrometeor growth

maximal reduction of the Raoult term (water activity) by organic mass and no surface tension reduction, the CF framework provides no reductions to the Raoult term, but a maximal reduction of the surface tension, since all organic material is assumed to reside in a film-like surface phase. The compressed film interfacial mechanism reduces $s_c$ and introduces a cusp in the Köhler curve once a particle exceeds the size required for complete film coverage (Fig. 1c, d).

Figure 2a, b shows vertical profiles of simulated cloud microphysics, for all three cases, during parcel ascension and cloud formation. By suppressing the critical supersaturation required for droplet activation, the CF model results in heightened CCN activity, increased number of growing hydrometeors and, therefore, an increased water vapour condensation sink as compared with BK simulations, reducing $s_{max}$ by 0.022, 0.044 and 0.088% for MA, HYY and NE, respectively (Fig. 2a). The corresponding enhancements in CDNC, denoted $\Delta_{CDNC}$, are 13, 26 and 145% (Fig. 2b). Increasing $s_{max}$ typically facilitates the activation of smaller aerosol particles, as described by Köhler theory, thereby enhancing CDNC for a given aerosol particle population. Here, however, CDNC is enhanced despite a decrease in $s_{max}$ due to larger reductions made to $s_c$ by the surface phase (Fig. 1c). The resulting changes in the smallest activated dry aerosol radius $r^*$ in the Aitken and accumulation modes are 5.6 and 3.9 nm (MA), 15.9 and 11.5 nm (HYY) and 9.3 and 2.9 nm (NE), respectively. In all cases, the CF droplet spectra at 200 m above the cloud base show a reduced mode size and increased

CDNC as compared with BK (Fig. 2c). While the difference in droplet size is only minor in the MA and HYY cases, it is significant for NE, owing to the combined effect of the organic fraction and aerosol size-distribution characteristics. These changes in cloud droplet spectra correspond to average changes in cloud optical thickness and albedo of ~4 and 3% (MA), 6 and 4% (HYY) and 36 and 25% (NE). Taking MA and HYY cases as global proxies for all oceanic and continental cloud systems, these cloud albedo changes could modify the short-wave cloud radiative effect over ocean and continental areas by as much as 11.5 and $-0.7$ W m$^{-2}$, respectively (see the Methods section for calculation details). While these values are undoubtedly a very coarse estimation, they suggest potentially significant effects of the description of the surface phenomena on the predicted climate implications of ACI, when compared against the estimated total indirect aerosol radiative forcing of $-1$ W m$^{-2}$ and total top-of-atmosphere short-wave cloud radiative effect (SW-CRE) of about[28] $-47$ W m$^{-2}$. These results highlight the strong sensitivity of the estimated aerosol forcing to global perturbations in CDNC and cloud droplet size.

**Exploration of the atmospheric parameter space.** While the results presented in Fig. 2 suggest potentially significant implications of the CF vs. BK treatment for CDNC, particularly on NE and HYY aerosol types, it is important to assess the magnitude of aerosol process effects simultaneously with variations in updraft velocities[20,29], organic fraction and aerosol size distribution

**Table 1 Overview of parcel model input parameters and output variables for bulk Köhler (BK) and compressed film (CF) models**

| Location | Mace head | Hyytiälä | Mace head |
|---|---|---|---|
| **Characteristic environment** | **Marine** | **Boreal forest** | **NUM event** |
| *Model input* | | | |
| Aitken mode number concentration $N_1$ [cm$^{-3}$] | 226 | 1110 | 2000 |
| Acc. mode number concentration $N_2$ [cm$^{-3}$] | 134 | 540 | 30 |
| Aitken mode geometric mean radius $R_1$ [nm] | 19.7 | 22.7 | 11.5 |
| Acc. mode geometric mean radius $R_2$ [nm] | 69.5 | 82.2 | 100 |
| Geom. standard deviation of Aitken mode $\sigma_{g,1}$ | 1.71 | 1.75 | 1.71 |
| Geom. standard deviation of accumulation mode $\sigma_{g,2}$ | 1.70 | 1.62 | 1.70 |
| Organic fraction in Aitken mode $F_{org,1}$ | 0.2 | 0.67 | 0.52 |
| Organic fraction in accumulation mode $F_{org,2}$ | 0.2 | 0.67 | 0.30 |
| Updraft velocity $w$ [m s$^{-1}$] | 0.32 | 0.32 | 0.32 |
| Minimum surface film thickness $\delta_{min}$ [nm] | 0.2 | 0.2 | 0.2 |
| Surface tension of organic $\gamma_{org}$ [mN m$^{-1}$] | 40 | 40 | 40 |
| *Model output* | | | |
| Cloud droplet number conc. CDNC$_{BK}$ [cm$^{-3}$] | 147 | 417 | 68 |
| Cloud droplet number conc. CDNC$_{CF}$ [cm$^{-3}$] | 166 | 513 | 165 |
| CDNC enhancement $\Delta_{CDNC}$ [%] | 13 | 23 | 145 |
| Maximum supersaturation $s_{max}^{BK}$ [%] | 0.272 | 0.183 | 0.407 |
| Maximum supersaturation $s_{max}^{CF}$ [%] | 0.250 | 0.138 | 0.319 |
| Smallest activated dry radius $\left(r_{1,BK}^*, r_{2,BK}^*\right)$ [nm] | 37.3, 32.8 | 65.6,61.2 | 34.8, 29.9 |
| Smallest activated dry radius $\left(r_{1,CF}^*, r_{2,CF}^*\right)$ [nm] | 31.7, 28.9 | 52.1, 52.1 | 25.5, 27.0 |
| Cloud optical thickness $\tau_{BK}$ | 5.6 | 7.9 | 4.3 |
| Cloud optical thickness $\tau_{CF}$ | 5.8 | 8.4 | 5.8 |
| Cloud-top albedo $\alpha_{BK}$ | 0.294 | 0.372 | 0.243 |
| Cloud-top albedo $\alpha_{CF}$ | 0.303 | 0.386 | 0.304 |

Optical properties are calculated assuming a cloud depth of 200 m, CDNC values are reported at cloud top

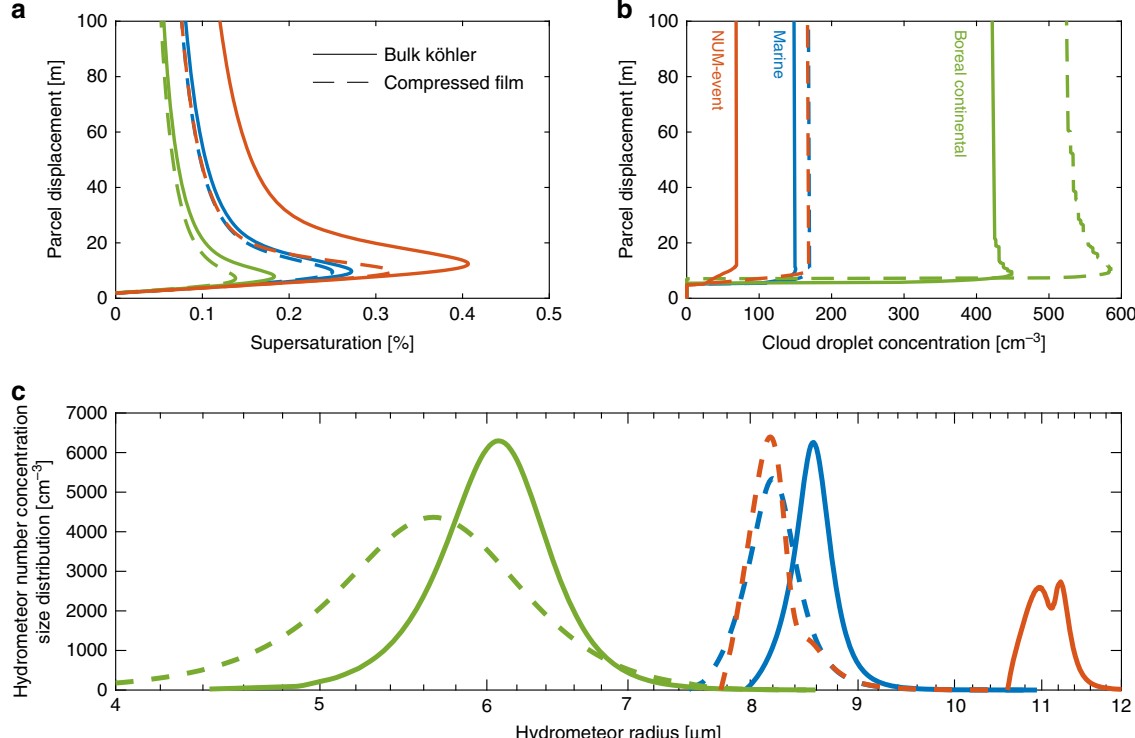

**Fig. 2** Simulated microphysics of cloud events on marine (MA, blue), boreal (HYY, green) and NUM-event (NE, orange) aerosol populations. Cloud-formation event simulations using bulk Köhler BK (solid lines) and approximate compressed film CF (dotted lines) models of cloud droplet activation with initial temperature $T = 280$ K, pressure $P = 98,000$ Pa, supersaturation $s = -0.1\%$ and fixed updraft velocity $w = 0.32$ ms$^{-1}$. Simulated (**a**) ambient parcel supersaturation and (**b**) cloud droplet number concentration during parcel ascent. **c** Simulated droplet size distribution at a parcel displacement 200 m above initialisation

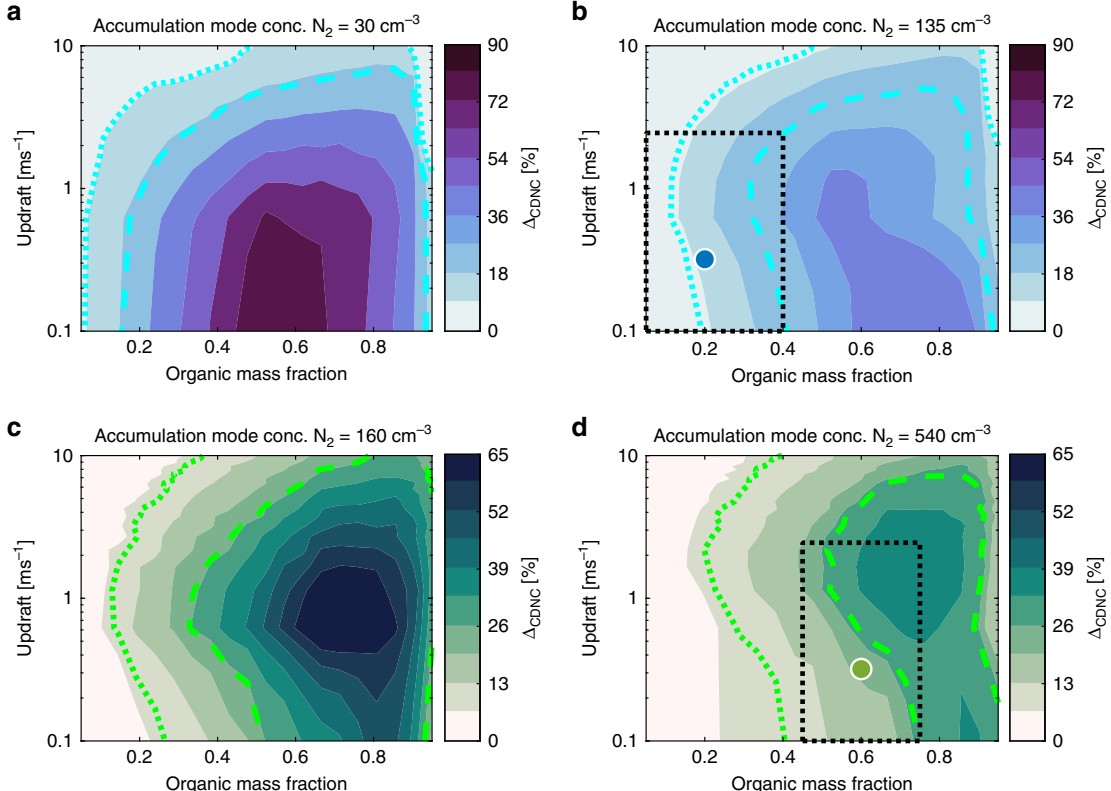

**Fig. 3** Cloud droplet number concentration response $\Delta_{CDNC}$ to the surface phase as a function of organic mass fraction and parcel updraft velocity. $\Delta_{CDNC}$ at 200 m above cloud base, as a function of the fixed updraft velocity and organic mass fraction for: **a, b** Marine (MA) simulations setting accumulation mode concentrations $N_2 = 30$ cm$^{-3}$ (corresponding to pristine marine accumulation mode, as observed during the NUM event[16]) and $N_2 = 135$ cm$^{-3}$, respectively. **c-d** Boreal (HYY) with $N_2 = 160$ cm$^{-3}$ and $N_2 = 540$ cm$^{-3}$, respectively. Blue and green markers indicate base MA and HYY simulation cases. Dotted boxes outline the regions of the parameter planes associated with typical environmental conditions (Table 2). Accumulation mode concentrations $N_2$ in panels **a** and **c** correspond to lower bounds for the studied environments[16,31], and those in **b** and **d** to average values. Dotted curves show the $\Delta_{CDNC}$ = 10% isoline. Dashed curves illustrate the environment specific $\Delta_\alpha$ = 5% isolines, i.e., $\Delta_{CDNC}$ = 25, and 28%, respectively for MA and HYY, determined from the Schwartz and Slingo approximation[64]

**Table 2 Parameter ranges bounding Latin hypercube sampled[38] random parameter combinations used for cloud parcel model input in assessing parametric uncertainties in model outputs**

| | Marine | | Boreal | |
|---|---|---|---|---|
| | Min | Max | Min | Max |
| Aitken mode number concentration $N_1$ [cm$^{-3}$] | 150 | 600 | 440 | 1780 |
| Acc. mode number concentration $N_2$ [cm$^{-3}$] | 60 | 250 | 160 | 920 |
| Aitken mode geometric mean radius $R_1$ [nm] | 15.5 | 23.5 | 14 | 31 |
| Acc. mode geometric mean radius $R_2$ [nm] | 70 | 100 | 66 | 98.5 |
| Geom. standard deviation of Aitken mode $\sigma_{g,1}$ | 1.40 | 1.80 | 1.56 | 1.95 |
| Geom. standard deviation of accumulation mode $\sigma_{g,2}$ | 1.40 | 1.80 | 1.46 | 1.78 |
| Organic fraction in Aitken mode $F_{org,1}$ | 0.05 | 0.40 | 0.60 | 0.71 |
| Organic fraction in accumulation mode $F_{org,2}$ | 0.05 | 0.40 | 0.60 | 0.71 |
| Updraft velocity $w$ [m s$^{-1}$] | 0.05 | 2.40 | 0.05 | 2.40 |
| Minimum surface film thickness $\delta_{min}$ [nm] | 0.16 | 0.30 | 0.16 | 0.30 |
| Surface tension of organic $\gamma_{org}$ [mN m$^{-1}$] | 30 | 50 | 30 | 50 |

throughout the input parameter space. To that end, Fig. 3 shows how the CDNC response to the surface phase depends on the updraft velocity $w$ and organic mass fraction $F_{org}$, for MA and HYY. NE is omitted here as a specific case with a well-defined[16] $F_{org}$, (see Supplementary Fig. 2 for NE updraft sensitivity). $w$ and $F_{org}$ perturbations to the MA and HYY cases indicate that total organic mass fractions exceeding ~0.2 and 0.3, for MA and HYY, respectively, result in $\Delta_{CDNC} > 10\%$ for all probed updraft velocities (Fig. 3b, d, Table 1). The $\Delta_{CDNC}$ peaks at $F_{org} \approx 0.6-0.7$ in

both cases, which is relevant for typical organic budgets in forested continental environments (dotted boxes in Fig. 3, corresponding to parameter ranges given in Table 2), but close to the upper limit of what has been observed for the marine aerosol during periods of peak biological activity[30]. In the MA case, peaks in $\Delta_{CDNC}$ were found for updrafts $w < 1$ m s$^{-1}$, and for the HYY case $1 < w < 3$ m s$^{-1}$. When $N_2$ is replaced with lower values associated with pristine maritime[16] and boreal environments[31], there is little change in the ranges of $F_{org}$ that facilitate $\Delta_{CDNC}$

**Table 3 Organic mass fraction and updraft criteria, established from Fig. 3b and d, yielding significant cloud microphysical responses to the surface phase**

| | Cloud microphysical susceptibility criteria | |
|---|---|---|
| Environment | $\Delta_{CDNC} > 10\%$ | $\Delta_\alpha > 5\%$ |
| Marine (MA) | $F_{org} > 0.2$ all $w$ | $0.35 < F_{org} < 0.95$ $w < 4.5$ ms$^{-1}$ |
| Boreal, Hyytiälä (HYY) | $F_{org} > 0.3$ all $w$ | $0.6 < F_{org} < 0.95$ $w < 6.0$ ms$^{-1}$ |

Criteria set on cloud albedo response to the surface $\Delta_\alpha = 5\%$ correspond to cloud droplet number concentration responses $\Delta_{CDNC} = 22$ and 28% for MA and HYY environments, respectively. Multiple criteria in a given cell must be satisfied simultaneously

**Table 4 Aerosol size-distribution criteria, established from Fig. 4a and d, yielding significant cloud microphysical responses to the surface phase**

| | Cloud microphysical susceptibility criteria | |
|---|---|---|
| Environment | $\Delta_{CDNC} > 10\%$ | $\Delta_\alpha > 5\%$ |
| Marine (MA) | $N_2 < aN_1^b + c$ | $100 < N_1 < 6000$ cm$^{-3}$ $N_2 < 150$ cm$^{-3}$ |
| Boreal, Hyytiälä (HYY) | all $N_1$ and $N_2$ | $N_2 < dN_1^e + f$ |

Criteria set on cloud albedo response to the surface $\Delta_\alpha = 5\%$ correspond to cloud droplet number concentration responses $\Delta_{CDNC} = 22$ and 28% for MA and HYY environments, respectively. Multiple criteria in a given cell must be satisfied simultaneously. Coefficients are: $a = 602$, $b = 0.0884$ and $c = -766$ ($R^2 = 0.9755$); $d = -832$, $e = -0.189$ and $f = 506$ ($R^2 = 0.8166$)

>10%, although the required $F_{org}$ values are below that of the characteristic MA simulation (Fig. 3a, c). This effect is also clearly demonstrated in the surface phase susceptibility of CDNC in the NE (Fig. 2), which contains the pristine maritime accumulation mode, $N_2 = 30$ cm$^{-3}$. While accumulation mode concentrations this low are probably extreme instances, these portions of the input parameter space illustrate how a reduction in the accumulation mode condensation sink can increase peak $\Delta_{CDNC}$ values as much as by 50 and 25%, for perturbed maritime and boreal environments, respectively. Similar subsets of the input parameter space pertaining to the CF model parameters and Aitken mode concentration are provided in Supplementary Figs. 3, 4 and discussed in Supplementary Note 1. While the $\Delta_{CDNC} > 10\%$ criterion is somewhat arbitrary and cloud optical properties will depend on absolute values of CDNC, it is chosen herein based on the sensitivity of SW-CRE estimates (see Methods) to the $\Delta_{CDNC}$ values reported here (Fig. 2). Constraints on $F_{org}$ and $w$ established by this criterion are summarised in Table 3, with additional, similarly derived constraints based on an alternative criterion set on albedo $\alpha$ enhancement of $\Delta_\alpha > 5\%$.

Model predictions of CDNC, cloud optical thickness and albedo with respect to changes in $N_1$ and $N_2$ (Fig. 4) share similarly shaped susceptibilities over the same domain and indicate that, besides requiring sufficient $F_{org}$, instances of large $\Delta_{CDNC}$ are linked to low $N_2$ and high $N_1$ for a fixed updraft $w = 0.32$ ms$^{-1}$ (Fig. 4a, d, Supplementary Fig. 3). The viability of $N_1$ and $N_2$ as reliable indicators of marine $\Delta_{CDNC}$, irrespective of aerosol modal size and width, was confirmed by analysis of the 2876 aerosol size distributions observed at Mace Head[32] during summer 2012 (see Fig. 4a scatter and Supplementary Note 2 for details). It should be cautioned that the Mace Head station is potentially influenced by anthropogenic sources[33], nevertheless simulations performed on the fitted concentrations in Fig. 4a are made using the MA composition to isolate aerosol size distribution dependencies. The time series of $\Delta_{CDNC}(t)$, corresponding to the scatter in Fig. 4, simulated on the observed size distributions has a median value of 10% (Supplementary Figs. 5–7). This indicates that the NE case is very atypical in its CDNC response of $\Delta_{CDNC} = 145\%$, and that the characteristic MA $\Delta_{CDNC} = 13\%$ simulation (Fig. 2), based on clean air masses, is not dissimilar to the potentially anthropogenically influenced air masses over this period on account of the size distribution.

A global sensitivity analysis, using the Sobol algorithm[25–27] (see Supplementary Note 2), in which all size distribution, compositional fractions, compressed film model and updraft parameters are varied simultaneously, authenticates these findings whilst also using a probability density function description of updraft velocities. Specifically, variance in absolute CDNC predictions is governed by aerosol size distribution and updraft parameters (Supplementary Figs. 8–9a, b, d, e), corroborating the

results from previous sensitivity analysis[34]. On the other hand, a similar analysis of the relative change $\Delta_{CDNC}$ (Supplementary Figs. 8–9c, f) yields a more uniform, interaction-dominated sensitivity landscape, particularly in the boreal case, in which compressed film parameters and the organic mass fraction also become important constituents of the total model variance. In summary, four key parameters, $N_1$, $N_2$, $w$ and $F_{org}$, are in minimum required to accurately capture the response of cloud microphysical properties to the organic surface phase. The molecular properties of the organic species (in this framework specifically the compressed film parameters $\gamma_{org}$ and $\delta_{min}$, see Table 1, Supplementary Figs. 4, 8, 9) will naturally proliferate in importance with increasing organic mass.

From Fig. 4, ranges of $N_1$ and $N_2$ yielding specific CDNC sensitivities can be determined for the studied conditions (assuming the compositions and updraft in Table 1). In the case of MA, $\Delta_{CDNC} > 10\%$ is achieved for $N_2 < aN_1^b + c$, a fitted a power law to the $\Delta_{CDNC} = 10\%$ contour in Fig. 4a (see Table 4 for the fitted coefficients, and Supplementary Note 4 and Fig. 10). Cloud albedo enhancements $\Delta_\alpha > 5\%$ are found for approximately $N_2 < 150$ cm$^{-3}$ and $100 < N_1 < 6000$ cm$^{-3}$ (Fig. 4c). It is clear from the specific positioning of the NE case in the ($N_1$, $N_2$) plane that its size distribution makes any potential cloud event particularly susceptible to the surface phase. For the HYY case, most of the ($N_1$, $N_2$) domain results in $\Delta_{CDNC} > 10\%$, whilst $\Delta_\alpha > 5\%$ is satisfied by a similarly fitted power law contour (see Table 4). Contours indicating significant microphysical susceptibility thresholds, in both MA and HYY cases, intersect with significant regions of the ($N_1$, $N_2$) plane associated with typical maritime and boreal aerosol concentrations (boxes, Fig. 4). It should be noted again that the choices of $\Delta_{CDNC} = 10\%$ and $\Delta_\alpha = 5\%$ as threshold values for cloud microphysical and optical responses are somewhat arbitrary and different choices for significant microphysical responses result in different concentration, organic fraction and updraft criteria, as illustrated by these two specifications. Nevertheless, these constraints can be used as a guide when deciding upon whether a detailed description of surface phenomena in a given environment for estimates of ACI is recommended (see also the typical ranges of aerosol concentrations in these environments, indicated by boxes in Fig. 4). The profile of the $\Delta_{CDNC}$ ($N_1$, $N_2$) surface is mimicked by cloud optical thickness and albedo enhancements, $\Delta_\tau$ and $\Delta_\alpha$, over the same domain (Fig. 4b, c, e, f). Peak values in $\Delta_{CDNC}$, $\Delta_\tau$ and $\Delta_\alpha$ are found to be 35 and 110%, 12 and 24% and 8 and 17%, for MA and HYY cases, respectively, which would all be expected to yield significant surface phase induced SW-CRE differentials[35]. The parametric sensitivities presented in Figs. 3, 4 and Supplementary Figs. 3, 4 support the conclusions obtained from the six representative cloud event simulations (Figs. 1, 2, Table 1): cloud formation in clean environments characterised by low

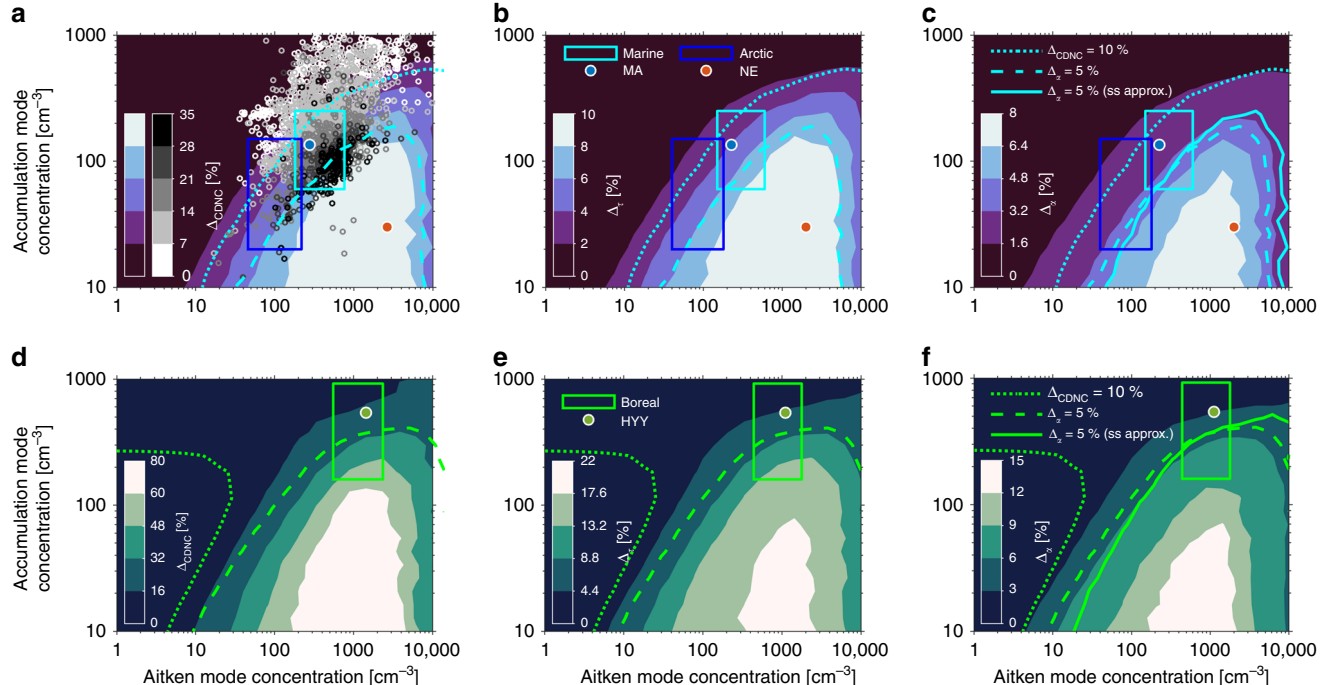

**Fig. 4** Cloud microphysical and optical property responses to the surface phase as a function of Aitken ($N_1$) and accumulation ($N_2$) mode concentrations. **a–c** Contour surfaces showing dependencies of cloud droplet number concentration, cloud optical depth and albedo enhancements $\Delta_{CDNC}$, $\Delta_\tau$ and $\Delta_\alpha$, respectively, assuming a cloud depth of 200 m, on $N_1$ and $N_2$ when updraft, chemical composition and other size-distribution parameters are fixed to their marine average (MA) values (Table 1, Fig. 1). Greyscale scatter shows simulated cloud droplet concentration enhancements $\Delta_{CDNC}$ on size distributions measured at Mace Head during summer 2012 assuming the MA composition and updraft used for the contour calculation. Red and blue scatter points show the MA and NUM-event aerosol concentrations in the ($N_1$, $N_2$) plane, and boxes show typical ranges applied to marine (cyan) and arctic marine (blue) aerosol concentrations[37,59]. **d–f** As in **a–c**, but for the Hyytiälä boreal simulation (HYY) and typical aerosol boreal concentration ranges. Dotted contours show the $\Delta_{CDNC} = 10\%$ isoline. Solid contours illustrate environment specific $\Delta_\alpha = 5\%$ isolines associated with $\Delta_{CDNC} = 22$ and 28% for MA and HYY, respectively, determined from the Schwartz and Slingo approximation[55]. Similarly, dashed contours indicate $\Delta_\alpha = 5\%$ isolines, but determined from explicit columnar integration of liquid water content and droplet effective radius and scattering asymmetry parameter. Constraints on aerosol concentrations necessary to exceed these microphysical metrics can be found in Table 4

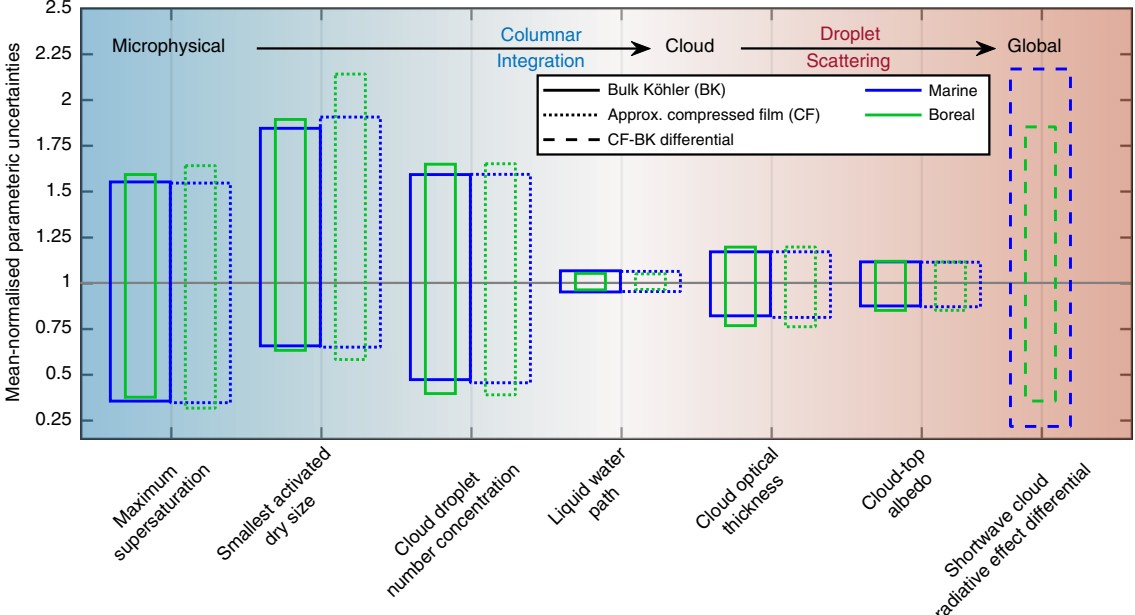

**Fig. 5** Normalised parametric uncertainties in parcel model simulated cloud microphysical and optical properties, and estimated surface phase induced short-wave cloud radiative effect (SW-CRE). Mean-normalised parametric uncertainties determined by dividing the 95th (top) and 5th (bottom) percentiles by mean values obtained from model output distributions resulting from simulations initialised with 5000 Latin hypercube sampled[38] parameter combinations from ranges in Table 2. A cloud depth of 200 m is assumed in all cases, and cloud droplet number concentrations statistics are made at cloud top. The SW-CRE differential is estimated by assuming fixed cloud fractions of 0.7 (marine) and 0.5 (boreal, land proxy)

accumulation mode concentrations but containing a source of ultrafine particles (e.g., via new particle formation) are most prone to a signal of surface-active organics in cloud droplet activation. For example, Arctic or Antarctic environments[36,37] could therefore be well-disposed to such influence in the presence of sufficient organic mass.

**Implications for aerosol–cloud–climate interactions**. Going from the micro- ($\Delta_{CDNC}$) to macrophysical optical ($\Delta_\alpha$) cloud responses results in diminished percentage change sensitivities (Fig. 4). Nevertheless, even minor changes in cloud-top albedo may have a significant impact on SW-CRE estimates, if widely applicable. As an example, assuming the MA simulation (Fig. 2) to be representative all marine clouds, a surface phase induced SW-CRE differential of $-1.5\,\text{W m}^{-2}$ follows a 3% albedo change assuming a cloud fraction of 0.7 over ocean and cloud depth of 200 m. By initialising the cloud model with input parameters (size distribution, $F_{org}$, $w$ and compressed film parameters, Table 2) randomly sampled using the Latin hypercube method[38], Fig. 5 illustrates how input parameter uncertainties translate into mean-normalised parcel model output variance across microphysical to bulk optical scales (see also Supplementary Figs. 11, 12 and Note 5). While the reduced sensitivity in the cloud liquid water path (LWP) is clear, the sensitivity remerges in the cloud optical properties, which depend explicitly on the cloud-top effective droplet radius. It should be noted, however, that the uncertainties presented in Fig. 5 do not account for perturbations in cloud depth and cloud fractions, which will likely result in quantitatively different estimates of uncertainty. SW-CRE estimates in particular will depend on explicit variation in cloud field features that are not captured by the adiabatic parcel model, though a simplistic dependence on fixed cloud fraction is given in the supplementary information (Supplementary Fig. 13). In terms of absolute model variances (Supplementary Figs. 11, 12), differentiating the CF and BK models from one another within the context of the natural variabilities expressed by the broad ranges given in Table 4 seems unlikely. Nevertheless, the combination of the small parametric uncertainty of the cloud optical properties and the large uncertainty in the resulting SW-CRE differential illustrates the inherent difficulty in distinguishing the signal from of microphysical processes from the observational data on cloud properties and accurately predicting the true climate impacts of detailed aerosol–cloud interaction processes.

## Discussion

The complex interactions between updraft, chemical composition and size-distribution shape in determining the role of surface-active species on cloud formation have been noted previously[20]. However, this is to our knowledge the first study probing the complete parameter space, outlining robust and quantitative constraints for conditions wherein the surface phase plays a significant role in ACI. These quantitative criteria encapsulate a diverse set of environmental conditions and provide guidance for determining instances where global climate models would benefit from chemically sophisticated, and often computationally more demanding, parameterisations of cloud formation. Our findings highlight the necessity of improved understanding of the chemical (e.g., emissions of specific chemical species and their reactions in the atmosphere) and microphysical processes (e.g., nucleation, condensation, coagulation and scavenging) that determine the shape of the aerosol number concentration size distribution, and their representation in climate models. Cloud formation in conditions characterised by clean accumulation modes and ultrafine particle sources are especially susceptible to

the surface activity and explain observed instances of high CCN activity such as the NE[16]. Furthermore, accurate knowledge of water vapour concentrations and its degree of saturation, the dynamic suppression of which we have here modelled with the approximate CF model for the first time, in different parts of the atmosphere are naturally a prerequisite of accurate predictions in ACI. By explicit simulation of surface phase suppressed ambient supersaturations, and the affect this has on hydrometeor growth rates, we have shown that cloud droplet sizes are reduced (Fig. 2c), despite the increase in critical sizes shown in Fig. 1c and previously reported[14] due to reduced growth rates and increased CCN activity. Dynamic suppression of ambient supersaturation by the surface phase in CF parcel model simulations of the NE also explains, at least in part, the discrepancy between the 145% CDNC enhancement reported herein and the observed[16] tenfold CCN concentration enhancement. The ambient supersaturation is also critically dependent upon the updraft velocity, herein shown to be one of the key drivers of surface phase dependent cloud microphysics and droplet activation, reiterating the need for its constraint on the GCM subgrid scale.

Besides the relatively high accumulation mode concentration, the small sensitivity displayed by the marine average case can in part be attributed to the low organic mass fraction (Figs. 1, 3 and 4). Whilst a modest prescription of organic mass is in-line with general modelling practices, its measurement at-large remains highly uncertain and instances of large sources of organics from marine biota have been reported[30]. Broader organic mass fraction perturbations (Fig. 3; Supplementary Fig. 9) indicate strong relative changes in CDNC, thereby advocating for improved organic aerosol budget prescriptions as has been noted previously[39,40]. More comprehensive observations on the exact identities, properties and amounts of atmospheric organic aerosol constituents are therefore warranted—particularly given the uncertainties related to the properties assumed for the representative surrogate systems. Furthermore, whilst the sensitivity of clouds to surface-active organics in typical boreal conditions was predicted to be moderate (at least compared with the NE), the results indicate an increase in the sensitivity at updraft velocities that can be present in convective regimes (Fig. 3). It is currently unknown if surface-active species are aerosol constituents in tropical convective storms, high organic mass concentrations, however, are known to be commonplace[41,42]. Collectively, these uncertainties highlight the need for long-term, size-resolved measurements of the molecular composition and properties of the organic aerosol fraction in different environments. We have focused on natural aerosol sources so as to provide some illumination on the importance and prevalence of chemically induced cloud microphysical susceptibilities in the modelling of preindustrial radiative forcings relevant for determination of the aerosol indirect effect. Therefore, similar analyses concentrating on anthropogenic aerosol, applying new estimates of marine organic abundance as they evolve and investigating surface-active organics in tropical convective clouds are required and should form the basis for future work. The assumption of complete solubility and insolubility of organic material in BK versus CF models makes the CDNC responses to the surface phase herein maximal, whilst the reality is probably somewhere in between. While recent studies indicate that surface tension reduction by organics does indeed prevail over the solute effect[14,16], laboratory studies with direct measurement of the surface tension reduction and surface composition of realistic atmospheric mixtures are also desirable[43].

In their totality, our findings further serve as a stark reminder that aerosol–cloud interactions are subject to complex non-linearities, and that the 'size vs. chemistry' narrative sometimes

perpetrated[44] is ill-posed, as their coupling in aerosol and cloud microphysics runs deep.

## Methods

**The adiabatic parcel model.** Here, we provide a brief overview of the adiabatic cloud–air parcel model, for a more extensive account the reader is referred to previous accounts[45,46] and references therein. The model simulates the ascent, expansion and cooling of an air parcel along with the size evolution of the hydrometeor population due to condensation/evaporation and co-evolution of liquid and vapour water mixing ratios. Mathematically, this takes the form of a system of $4 + n_t n_b$ coupled ordinary differential equations where $n_b$ is the number of size classes composing each of the $n_t$ aerosol type (as defined by distinct internal mixtures) size distributions that constitute external mixtures; each size class in each aerosol type has an associated condensation/evaporation differential equation. The four additional differential equations constitute the evolution of parcel temperature $T$, pressure $P$, supersaturation $s$ and altitude $z$. Initial conditions are set to $T_0 = 280$ K, $P_0 = 98000$ Pa and $s_0 = -1\%$ to characterise conditions immediately below cloud base and are constant across all simulations performed. Specification of $z$ is arbitrary, and therefore all model output is shown in terms of parcel displacement. To mimic the conceptual framework of droplet activation parameterisations employed in GCMs, all simulations were carried out using constant updraft velocities $w$, as opposed to buoyancy-derived updraft velocities, and therefore the rate of ascent is simply the initial updraft

$$\frac{dz}{dt} = w. \tag{1}$$

The updraft velocity is used in determining the rate of adiabatic cooling of the parcel

$$\frac{dT}{dt} = \frac{L_e}{c_{p,a}}\left(\frac{dw_l}{dt}\right) - \frac{gw}{c_{p,a}}, \tag{2}$$

where $L_e$ is the latent heat of water evaporation, $c_{p,a} = 1.006$ kJ kg$^{-1}$ K$^{-1}$ the specific heat capacity of air, $w_l$ the liquid water content of the parcel and $g = 9.81$ m s$^{-2}$ the acceleration of gravity. The second term of the right hand side describes the dry adiabatic lapse, and the first term the correction due to latent heat released into, or depleted from, the gas phase due to the condensation or evaporation. Evolution of the parcel pressure is determined from the hydrostatic balance:

$$\frac{dP}{dt} = -\frac{gw}{R_a T}P, \tag{3}$$

where $R_\alpha = 287.058$ J kg$^{-1}$ K$^{-1}$ is the specific gas constant of air. Evolution of the supersaturation is dictated by the updraft source and condensation sink, and can be expressed as

$$\frac{ds}{dt} = -\frac{R_v P}{R_a e_s}\left(\frac{dw_l}{dt}\right) - (1+s)\left[\frac{L_e}{R_a T^2}\left(\frac{dT}{dt}\right) + \frac{g}{R_a T}w\right], \tag{4}$$

where $R_v = 461$ J kg$^{-1}$ K$^{-1}$ is the specific gas constant of water vapour, $e_s$ the saturation vapour pressure of water given by a sixth-order temperature polynomial[47]

$$e_s(x) = \sum_{i=1}^{6} a_i x^i, \tag{5}$$

where $x = T - 273.15$ K, and $a_0 = 6.108$, $a_1 = 4.437 \times 10^{-1}$, $a_2 = 1.429 \times 10^{-2}$, $a_3 = 2.650 \times 10^{-4}$, $a_4 = 3.031 \times 10^{-6}$, $a_5 = 2.034 \times 10^{-8}$ and $a_6 = 6.137 \times 10^{-11}$.

To solve the above system of ordinary differential equations the condensation sink must be evaluated at each time step $i$ of duration $\Delta t = 0.01$ s

$$\left(\frac{dw_l}{dt}\right)_i = \frac{\text{LWC}_i - \text{LWC}_{i-1}}{\Delta t}, \tag{6}$$

where the liquid water content LWC in units of $g$, $g_{\text{air}}{}^{-1}$ is determined from the wet and dry radii

$$\text{LWC} = \frac{4\pi\rho_w}{3\rho_a}\sum_{k=1}^{n_t}\sum_{j=1}^{n_b} n_{jk}\left(r_{jk}^3 - r_{jk,\text{dry}}^3\right), \tag{7}$$

and $\rho_a$ is the density of air calculated from the ideal gas equation. Dry radii $r_{jk,\text{dry}}$ are 400 size classes logarithmically distributed between 2 and 500 nm for MA and HYY simulations, and 10.5 and 251 nm for NE for consistency with the observational data set used. Wet radii are calculated using analytical solutions to the linearised form[48] of the condensation equations describing heat and vapour diffusion

$$r_{jk}\frac{dr_{jk}}{dt} = \frac{(S - S_{\text{eq},jk})}{\frac{\rho_w R_v T}{D_v^* e_s} + \frac{L_e \rho_w}{kT}\left(\frac{L_e}{R_v T} - 1\right)}, \tag{8}$$

$k = 4.2 \times 10^{-3}\,(1.0456 + 0.017\,T)$ is the thermal conductivity[49] measured in J m$^{-1}$ s$^{-1}$ K$^{-1}$, and $S_{\text{eq}}$ the saturation ratio at the particle surface as evaluated by the bulk Köhler or compressed film models (see the next section). $D_v^*$ is the

particle size-dependent water vapour diffusivity[49] measured in m$^2$ s$^{-1}$,

$$D_{v,jk}^* = \frac{D_v}{\left(1 + \frac{D_v}{\alpha r_{jk}}\sqrt{\frac{2\pi M_w}{RT}}\right)}, \tag{9}$$

where $\alpha$ is the dimensionless water vapour mass accommodation coefficient, $M_w = 18.016$ g mol$^{-1}$ is the molecular mass of water and $D_v$ the size-independent diffusivity[50],

$$D_v = 0.211 \times 10^{-4}\left(\frac{P_0}{P}\right)\left(\frac{T}{T_0}\right)^{1.94}. \tag{10}$$

$P_0 = 1$ atm and $T_0 = 273.15$ K being the reference pressure and temperature. Since the number concentration of aerosol particles in each size-bin remains constant, the cloud droplet number concentration (CDNC) can be derived by integrating the continuous dry aerosol size-distribution function $n_k(r_{a,\,k})$ from the smallest activated dry size class $r_k^*$ (defined as the dry radius of the smallest activated aerosol of type $k$ such that $r_{jk} > r_{jk,c}$) to infinity,

$$\begin{aligned}\text{CDNC} &= \sum_{k=1}^{n_t}\int_{r_k^*}^{\infty} d\ln(\text{r})n(\ln(r)) \\ &= \frac{1}{2}\sum_{k=1}^{n_t} N_k\left(2 - \text{ERFC}\left[\frac{\ln(R_k) - \ln(r_k^*)}{\sqrt{2}\ln(\text{GSD}_k)}\right]\right),\end{aligned} \tag{11}$$

where ERFC[$x$] is the complementary error function evaluated on $x$, and $N_k$, $R_k$ and GSD$_k$, are the mode concentration, radius and geometric standard deviation of the $k$th aerosol type.

**Modelling the organic aerosol surface phase.** Assuming ideality and a dilute aqueous phase, Köhler theory expresses the saturation vapour pressure ratio $S_{eq}$ at the surface of a wetted particle in terms of its dependency upon $A$ and $B$ coefficients, which characterise the surface curvature (Kelvin term) and solute (Raoult term) effects, together in the Köhler equation[51]

$$S_{\text{eq}} = e^{\left(\frac{A}{r} - \frac{B}{r^3}\right)},\ A = \frac{2\nu\gamma}{RT},\ B = \frac{3n_s\nu}{4\pi}, \tag{12}$$

$R = 8.314$ J mol$^{-1}$ K$^{-1}$ is the universal gas constant, $\nu$ the molar volume of the particle liquid, $\gamma$ the surface tension of the particle and $r$ its wet radius. Herein, the number of moles of solute in the bulk aqueous phase $n_s$ is composed of both dissolved inorganic and organic matter, both assumed to be completely soluble in the bulk Köhler scheme. Component species' molar volumes $v_i$ and input volume fractions $f_i$ are used to determine the total number of soluble moles

$$n_s = \frac{4\pi r_{\text{dry}}}{3}\sum_{i=1}^{N_s}\frac{\phi_i f_i}{\nu_i}, \tag{13}$$

where $N_s$ is the total number of species, $\phi_i$ their ionic dissociation and the $f_i$ are determined from the respective desired mass fractions $F_i$ assuming volume additivity

$$f_i = \frac{\frac{F_i}{\rho_i}}{\sum_{j=1}^{N_s}\frac{F_j}{\rho_j}}, \tag{14}$$

where $\rho_i$ is the density of compound $i$.

This formulation that assumes a single bulk homogeneously mixed aqueous phase and a size-independent surface tension of a pure water drop $\gamma_w = 72.8$ mN m$^{-1}$, is bulk Köhler theory (BK). The peak of the bulk Köhler curve referred to by the critical supersaturation $s_c$ (defining thermodynamic CCN activation potential) and critical radius $r_c$ (defining kinetically realised cloud droplet formation) can be calculated by finding the stationary point,

$$r_c = \left(\frac{3B}{4A}\right)^{1/2},\ s_c = \left(\frac{4A^3}{27B}\right)^{1/2}. \tag{15}$$

Here, we have embedded an approximation[16] to the compressed film (CF) model[14] into the parcel model to determine $S_{\text{eq}}$ in the presence of organic surface-active components, which assumes a maximal effect with respect to the Raoult term in the Köhler equation, since all organic mass is assumed completely insoluble and residing in a film-like surface phase. In this view, the organic mass makes no contribution to $n_s$, or therefore the bulk water activity, in the above formulation. As a result of the organic surface phase, surface tension is depressed and modelled in terms of the fractional surface coverage

$$\varepsilon = \min\left[\frac{V_{\text{org}}}{V_\delta}, 1\right], \tag{16}$$

where $V_\delta$ is the volume of the surface phase associated with a minimum film thickness $\delta_{\min}$, typically on the order of the size of a single molecule, assumed to be $\delta_{\min} = 0.2$ nm herein unless otherwise stated. $V_{\text{org}}$ is the volume of the organic surface phase, determined from the molar volume of the organic compound. From the fractional surface coverage, the surface tension is derived from a surface coverage weighting of the pure organic $\gamma_{\text{org}}$ and pure water $\gamma_w$ surface tension

values

$$\gamma = \varepsilon\gamma_{org} + (1 - \varepsilon)\gamma_w, \tag{17}$$

where $\gamma_{org} = 40$ mN m$^{-1}$ unless otherwise stated. With the surface tension and bulk concentrations determined, $S_{eq}$ can be computed in the compressed film framework and substituted into the condensation equations above. Since for the CF model analytical expressions for $r_c$ and $s_c$ are not currently available, a simple max-function is used in a prior calculation of CF Köhler curves.

**Simulation setup.** Aerosol particles in the marine case, denoted MA, are prescribed by the average of size distributions measured at Mace Head, Ireland[52] on account of its status as a remote location subject to influxes of maritime air masses[53]. While instances of the organic mass fraction $F_{org}$ as high as 0.63 have been reported during peak phytoplankton bloom periods[30], here a more conservative value of 0.2 for both Aitken and accumulation modes is chosen to reflect annual average values observed using the Aerosol Mass Spectrometer (AMS)[53]. The remaining aerosol mass is taken to be ammonium sulphate and sodium chloride—as a surrogate for the inorganic sea spray[54]—in the Aitken and accumulation modes, respectively. In representing the chemical properties of the MA organic fraction with a single surrogate compound, the molar mass $M_{org} = 256.4$ g mol$^{-1}$ and density $\rho_{org} = 0.852$ g cm$^{-3}$ of palmitic acid[15,55] are chosen.

The remote continental case is characterised using aerosol size distributions measured at the SMEAR II station in Hyytiälä, Finland[31], further denoted HYY, and prescribing $F_{org} = 0.60$ based on available (mostly spring, summer and early fall) AMS measurements[56], for both Aitken and accumulation modes. To chemically characterise the HYY Aitken and accumulation mode organic fractions, average properties of SOA surrogate systems 2 and 1, respectively, previously reported[16] are assumed. Since these systems are thought to be representative of SOA from alpha-pinene ozonolysis, they may be deemed suitable proxies of also the boreal type SOA[39,57]. Specifically, the dimer compounds, assumed to be present in the Aitken mode, are thought to represent low-volatility, multifunctional organic compounds; the average properties of which are $M_{org} = 368.4$ g mol$^{-1}$ and $\rho_{org} = 1.2$ g cm$^{-3}$. The more chemically diverse SOA surrogate system 1, assumed to constitute the organic fraction of the accumulation mode, has average properties $M_{org} = 190$ g mol$^{-1}$ and $\rho_{org} = 1.24$ g cm$^{-3}$.

The NE aerosol size distribution and organic and inorganic mass fraction composition were taken to be those reported at Mace Head, Ireland[16], and the organic components of each aerosol type are taken to be the SOA surrogates also used in HYY—although the exact composition of the NUM event is not known.

These cases are of course coarse in their representation of maritime and boreal aerosol populations in relation to the large degree of spatiotemporal variability, potential size-dependent composition of the ultrafine aerosol, as AMS data are generally dominated by the large end of the size-distribution probed, and the expected complexity of realistic atmospheric organic aerosol (and even inorganic sea spray[54]). Nevertheless, we employ them first as specific model inputs for representative cloud formation simulations, and second as a basis on which to explore the input parameter space associated with natural variability in aerosol physicochemical properties.

For the CF model, a minimum film thickness of 0.2 nm and pure organic surface tension of 40 mNm$^{-1}$ were assumed for all the cases (Table 1), while extensive literature on these values for the representative mixtures is missing. A typical fixed updraft velocity of $w = 0.32$ m s$^{-1}$ (see Supplementary Fig. 2 and Note 6) was used throughout, unless otherwise stated, to isolate the effects of the aerosol population and phase separation of the organic mass. For each case (MA, HYY and NE), an adiabatic cloud event was simulated twice, first with a single bulk phase Köhler theory description of cloud droplet growth (denoted BK), assuming complete dissolution of the organics, and second with an approximation[16] to the compressed film model[14] further denoted CF (Fig. 1; Supplementary Fig. 1). Whilst the BK approach represents a limiting case with maximal reduction of the Raoult term (water activity) by organic mass and no surface tension reduction, the CF framework provides no reductions to the Raoult term, but a maximal reduction of the surface tension, since all organic material is assumed to reside in a film-like surface phase. The compressed film interfacial mechanism reduces $s_c$ and introduces a cusp in the Köhler curve once a particle exceeds the size required for complete film coverage (Fig. 1c, d).

The parameter value ranges used for the sensitivity and parametric uncertainty analyses for MA and HYY are listed in Table 2. For MA, suitable ranges of the organic fraction are chosen based on AMS measurements reported to originate from arctic, polar and tropical maritime sources[53], while the scope of size-distribution parameters is taken from those reported by Heintzenberg et al.[58]. Ranges in organic fraction for the boreal case HYY are obtained by quadrature uncertainty combination of different components of the organic fraction determined from statistical analysis of measurements made in Hyytiälä[56]. The corresponding size-distribution ranges are derived from standard deviations reported for continental air masses in Hyytiälä[31].

**Cloud optical properties and radiative effect estimates.** To relate the particle size distribution to cloud optical properties, the total liquid water content along the column of ascent, termed the liquid water path LWP and measured in units of g m$^{-2}$, must be determined. From first principles, the LWP is readily computed by

integration over an assumed cloud depth $z_c$

$$\text{LWP} = \int_0^{z_c} dz\rho_a(z)\text{LWC}(z). \tag{18}$$

For calculation of optical properties herein, we assume $z_c = 200$ m throughout as a typical stratocumulus cloud depth. Whilst simplifying expressions for the integral can be made for particular (limiting) cases[59,60] of cloud LWC profiles, given the computational efficiency of the model and the broad parameter ranges probed throughout the analysis, performing numerical integration by default is deemed judicious. The dimensionless liquid cloud optical thickness $\tau$ can be derived[59] from the LWP and cloud-top effective radius of the droplet population $r_e$,

$$\tau = \frac{3\text{LWP}}{2\rho_w r_e}, \tag{19}$$

where the effective radius is defined by the ratio of the third and second moments of the cloud droplet spectrum[61], $r_e = r^3/r^2$. From the optical thickness, the cloud albedo $\alpha$ may be determined subject to the two-stream approximation[62],

$$\alpha = \frac{(1-g)\tau}{2 + (1-g)\tau}, \tag{20}$$

where the dimensionless asymmetry parameter $g$, a property of the droplets, used to characterise the radiative transfer of energy within the droplet is typically assumed to be[63] ~0.85, indicative of a forward scattering. In addition, absolute changes in cloud albedo $\Delta(\alpha)$ can be approximated from CDNC changes[64] $\Delta(\alpha) = 0.075\times\ln(\text{CDNC}_{CF}/\text{CDNC}_{BK})$, where here $\text{CDNC}_{BK}$ and $\text{CDNC}_{CF}$ are CDNC corresponding to BK and CF models. Rearranging and using $\text{CDNC}_{BK} = 139$, 185 and 410 cm$^{-3}$, for MH, MA and HYY, respectively (Table 1) indicates that CDNC enhancements of $\Delta_{CDNC} = 4$ (22, 49), 4 (23, 54) and 5 (29, 66) are required for 1 (5, 10)% increases in cloud albedo relative to BK albedo estimates obtained from the parcel model.

If the surface phase induces an absolute albedo difference $\Delta(\alpha) = \alpha_{CF} - \alpha_{BK}$ (Eq. (20)) for a given aerosol type, then an associated short-wave cloud radiative effect (SW-CRE) differential can be estimate[65],

$$\Delta(F) = -\varepsilon_s\varepsilon_c \frac{F_0\Delta(\alpha)}{4}, \tag{21}$$

where $F_0 = 1370$ W m$^{-2}$ is the solar irradiance received by the top of the atmosphere, and $\varepsilon_s$ and $\varepsilon_c$ are the fractional surface coverage of the Earth made by the given aerosol type and of cloud coverage prevailing over that aerosol type, respectively. Using input parameters in Table 1 and assuming a cloud depth of 200 m, absolute changes in cloud albedo are found to be $\Delta(\alpha) = 0.009$ and e $\Delta(\alpha) = 0.014$ for MA and HYY, respectively. Taking MA and HYY cases as proxies for ocean and land aerosol populations, $\varepsilon_s = 0.7$ and 0.3 and average cloud fraction[66] $\varepsilon_c = 0.7$ and 0.5, this simplified formulation yields respective SW-CRE differentials of $\Delta(F)_{ocean} \approx -1.5$ W m$^{-2}$ and $\Delta(F)_{land} \approx -0.7$ W m$^{-2}$. Such a calculation is a dramatic simplification in its assumption of an ever-present organic surface phase, and globally uniform cloud depth, and neglect of feedback effects. In that respect, these numbers should be considered order of magnitude estimates, nevertheless, when combined they are on the order of that of total aerosol forcing estimates[4], $F_{tot} \sim -1$ W m$^{-2}$.

## Data availability
The data used to produce Figs. 1–5 are available from the Bolin database (https://bolin.su.se/data/lowe-2019) and/or upon request from the authors. The observational data in Fig. 4 has been acquired from the EBAS database (www.ebas.nilu.no).

## Code availability
Plotting, data analysis and simulation setup scripts are available at https://github.com/SamJLowe/NatComms_OrgSurfaceClouds.

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

## Acknowledgements

This work was part of the AtmoRemove project funded by the Knut and Alice Wallenberg foundation (project number 2015.0162). K.R.W.'s contribution to this work is supported by the Condensed Phase and Interfacial Molecular Science Program, in the Chemical Sciences Geosciences and Biosciences Division of the Office of Basic Energy Sciences of the U.S. Department of Energy under Contract No. DE-AC02-05CH11231. Drs. Douglas Nilsson and Claudia Mohr are gratefully acknowledged for their help in constructing the representative cases for the MA and HYY environments. Prof. Annica Ekman is gratefully acknowledged for her help in deriving short-wave cloud radiative effect estimates. D.G.P. would like to express his gratitude to Dr. Geert-Jan Roelofs for providing him with the cloud parcel model that was developed and used for this study. Open access funding provided by Stockholm University.

## Author contributions

S.J.L., I.R., D.G.P., K.W. and J.D. conceptualised the study. S.J.L., D.G.P. and I.R. designed the simulations and developed the methodology. S.J.L. developed the parcel model for inclusion of the approximate compressed film mechanism and conducted the simulations. D.G.P. developed the framework for coupling a parcel model to the Sobol method. S.J.L., I.R., D.G.P., K.W., J.D. and D.T. participated in analysis of the results and commented on the paper. S.J.L. and I.R. wrote the paper.

## Competing interests

The authors declare no competing interests.
