## [Peer Review File · Nature Communications]

Reviewers' comments:

Reviewer #1 (Remarks to the Author):

The paper uses bulk Kohler and compressed film models in combination with an adiabatic parcel model to investigate the sensitivity of CDNC to aerosol size distribution and particle composition – specifically mass fraction and composition of organics. As the authors state, the results provide a basis for improving climate model simulations of ACI through the incorporation of molecular-level phenomena. In addition, the results should help guide collection of the most relevant observational data for a given region/aerosol type. The emphasis on the need for measurements that help constrain the organic aerosol budget is not new but the paper provides a strong case for the application of these measurements in reducing uncertainties in simulations of ACI. My comments primarily concern assumptions made about the composition of the three different aerosol types. I would have liked more detail on what went into the assumptions. This and other specific concerns are listed below.

Lines 118 – 120: This appears to be the only place in the manuscript and supplementary material where sources of the assumed composition for each aerosol type are described. Please provide more details about the assumed composition as a function of particle size. Why are the Aitken and accumulation mode compositions different for MH but not HYY and MA as shown in Figure 1b? This seems to contradict the information given in Table 1 where the accumulation mode organic fraction for MA and HYY are listed as zero. What is the choice of organic component based on for each aerosol type? Is the surface tension for each of these organic components the same as indicated by an assumed uniform surface tension value of 40 (Table 1)?

Lines 134 to 137: It is acknowledged that these changes in the activated particle size are small. Can some uncertainty be assigned to the changes in r^* ?

Lines 143 – 144: Please state explicitly why the change in droplet size and CDNC for the CF simulations are largest for MH. Presumably it has to do with composition since the organic mass fractions at HYY are larger.

Figure 1b: Different colors should be used so that it is easier to tell which compound is which. Also – change to “b) ASSUMED chemical composition...”

Line 723: Shouldn't this be accumulation (N_2)?

Lines 229 – 232: What are the upper thresholds in N_2 and N_1 (300 and 1190 cm^{-3}) as representative of unpolluted air masses based on?

Figure 3: I found this figure to be exceedingly complicated because of the amount of information it contains. I am not sure how to simplify it, though, given the length constraints of the journal.

Reviewer #2 (Remarks to the Author):

Overall, this is an interesting study addressing an important topic in a new way, but more details are required before this can be published. Specific comments follow.

The authors need to provide more details regarding how their results depend on the chosen condensed film parameters. Why 0.2 nm? The studies cited show a diversity of behavior, depending on the specific organic considered. It doesn't seem that differences in CF film parameters are considered in the sensitivity analysis. Is there a reason why not?

More information is required regarding how the particle composition was assumed to vary with size. It is certain that the composition of the Aitken mode particles differs from the accumulation. Only for the coastal case do the authors seem to account for this. Also, they assume primarily

NaCl for the marine case. But the Aitken mode is unlikely to be NaCl, instead more likely to be sulfate (perhaps even quite acidic, as in Ovadnevaite). Does this matter? And why is HYY succinic acid while MH is an ester dimer? (Note: the citation of Ovadnevaite is insufficient, as they didn't actually detect these molecules. This was a guessed at part of the interpretation, as best I can tell.) Also, why are they using Fulvic acid for the MA particles, but citing a paper that does not use fulvic acid and, in fact, suggests that fatty acids are not especially good at impacting activation. None of this seem particularly justified, but perhaps more important it is not clear how it matters since it seems that the CF parameters used are simply constants with $\delta = 0.2$ nm. So, does the variability in the organic composition only impact the Raoult term for the BK approach? This lack of clarity regarding composition is my main concern.

Relating to the previous comment, it is not totally clear to me that the authors have really covered a sufficient parameter space to make concrete conclusions regarding sensitivity. They have most certainly done quite a bit and their results provide important guidance. But the lack of exploration involving composition (of both the organics and the inorganics) is a limitation that is not sufficiently addressed.

I strongly suggest that the authors update the abstract to be a bit more specific to the study. When they state things like "The ACI" this can mean many things.

L116: Does the BK case really represent the "maximum reduction of the Raoult effect"? Max reduction in what? Won't the Raoult effect be maximized in the BK case?

L134: Would be clearer to state "The resulting decrease in the size of the activated particles for the Aitken or accumulation mode particles at each site is"

L141: The details regarding how the Ovadnevaite case are unclear. The authors need to be much clearer about how this is done (size-dependent composition? Dry size distribution)?

L151: How is "Significance" determined?

L173: I feel as if the discussion around this section could benefit from mention of Dusek et al. (Science, 2006), who conclude that "size matters more than chemistry."

L248: Suggest deleting the word "convoluted."

It is generally indicated that the authors are using the "compressed film" model. But they are actually using an approximation of the compressed film model (as stated in Ruehl et al.). This should be clarified.

Reviewer #3 (Remarks to the Author):

Review of 'Constraining the role of surface active organics in cloud formation'

Aerosols and clouds cause the largest uncertainty to estimates and interpretations of the Earth's changing energy budget (IPCC, 2013). The complexity, however, associated with elucidating the specific organic matter effects is unrivalled compared to other pollutants e.g. inorganic compounds. Carbonaceous (organic matter and black carbon) are not even properly represented in the present-day models. Especially the organic effect on surface tension that has been observed to be important in cloud activation, but is not currently accounted in regional or global climate models. The use of static pure water surface tension, despite actual composition information, is still predominant in these models. What complicates matters even more is that there is no unifying theory for all types of situations, or, in other words, there is no one size fits all model and the classical κ theory with water surface tension is suitable for numerous of applications, but neglecting the effects of surface tension causes significant errors in others. Hence, it is important

to define situations when accounting for surface tension reduction is crucial and when simple kappa theory is justifiable. An attempt to present the quantitative criteria for when surface effects should be accounted for in the global models, in this manuscript, is, thus, highly appreciated and timely. However, I see some significant drawbacks for case selection on which these criteria are based.

The main issue is with a representativeness of selected reference cases/studies for conditions representing selected environments. Very specific cases constrained to one location (MH, HYY) or old paper for MA that does not cover other aspects of marine aerosol (specifically organic fraction in sea spray aerosol) are not really supporting the ambitious aim of this manuscript. As I noted above, the task is very timely and needed, but has to be based on more robust environmental description.

Heintzenberg et al. paper is conceptually old and do not even include organic matter in the marine aerosol composition. Assumption that sea salt and nssSO₄ with MSA are the main components of the marine aerosol might have been the knowledge of the time, but this manuscript focuses on surface tension effects by organics, so it is essential to account for new knowledge on marine organics, both primary (sea spray) and secondary. Moreover, Heintzenberg et al. paper refers to marine environment as being strongly affected by anthropogenic influences, which again was more true at the time, but several recent sulphur isotope papers show that the anthropogenic contamination has significantly reduced over the oceans due to major reduction in sulfur pollution over the continents itself. Maybe, the NASA-sponsored Atmospheric Tomography Mission (ATom) campaigns and papers can provide better information on marine aerosol size distributions and compositions or use coastal stations data filtered for pristine marine conditions can help here.

Similarly, Mace Head data are referred to as coastal with terrestrial influence, which is correct in general, but the cases presented in this manuscript correspond to exclusively marine aerosol sampled at Mace Head. E.g. Line 60 and elsewhere, NUM event in Ovadnevaite et al. is referred to as coastal location with terrestrial influence, while authors of the original paper refer to pristine marine air masses (Black carbon mass concentrations <5 ng m⁻³ confirm measurements within a pristine marine air mass). While a general classification of Mace Head as coastal location with terrestrial influence is justifiable when criteria of marine air masses are not applied, the reference in this case is not appropriate without arguing why the intended meaning of the authors is not accepted here. Similarly in the conceptual diagram (Figure S1), Mace Head is presented as coastal site with terrestrial influence, but then NUM case from Ovadnevaite et al. is not appropriate for general example.

On the same note, Mace Head, as being coastal location, would certainly have an impact from sea spray, but this is not reflected in Figure 1 b) or anywhere else in the manuscript. Hence, the confusion again, does Mace Head case refer only to particle nucleation mode, as shown in Ovadnevaite et al., or was it meant to be a general representation of the coastal area. If the latter, sea spray particles should also be included, especially in the accumulation mode. Moreover, the selection of sodium chloride as a proxy for sea salt for marine case is not reflecting the current knowledge of the sea spray as it is known that solubility, hygroscopicity and, thus, cloud activation for NaCl differ from that of sea salt (Zieger et al., 2017). Finally, it is not clear whether O'Dowd et al. 2004 reference was used for coastal particle composition or marine as it shows much higher OM contributions than it is currently presented in Figure 1 for marine case;

More recent studies show high fraction of marine organics in sea spray (O'Dowd et al., 2004; Facchini et al., 2008; Ovadnevaite et al., 2011; Rinaldi et al., 2013) and its global importance (Spracklen et al., 2008; Gantt et al., 2012), what effect would that have on the conclusions here for MA case, especially in lines 145-146?

Elaborate on the conclusion in lines 143-144 stating that 'The implications of surface activity for cloud microphysics are hence more significant for the coastal instance, despite the larger organic mass fraction at HYY';

Another major problem is with differentiation and proper use of Compressed Film (CF) vs liquid-liquid phase separation (LLPS) models, here, they are combined into one and named as CF despite

being principally different:

Ovadnevaite et al. paper uses an equilibrium gas–particle partitioning and liquid–liquid phase separation (LLPS) model with variable surface tension as opposed to compressed film model and these are not interchangeable. So which inputs did you use – based on CF or LLPS?

Figure 1 d) is better suited to illustrate an equilibrium gas–particle partitioning and liquid–liquid phase separation (LLPS) model rather than compressed film model as the latter refers to continuous droplet surface coverage by OM with only the film thickness varying with droplet growth (the compressed film model predicts that a particle will reach Sc when the surface film decreases in thickness to the point that individual molecules begin to separate; i.e., a 2D phase transition occurs, and the surface tension no longer varies with increasing D_{wet}) (Ruehl et al., 2016). Be specific what are you refereeing to when talking about compressed film model. Also, CF model does not properly represent small particle effect (Davies et al., 2019), was this accounted in making conclusions for NUM particles (lines 224-226)?

Do you state that compressed film model results in smaller droplet diameter here (Lines 138-140)? Explain how it compares with CF model in (Ruehl et al., 2016) and LLPS in (Ovadnevaite et al., 2017) where they show an increase in wet diameter?

Finally, it is claimed that robust and quantitative constraints for conditions wherein the surface phase plays a significant role in ACI are presented, but it is not entirely clear from the manuscript what those conditions are. While reader might be able to draw them from figures and supplementary tables presented, the ambitious aim requires listing them clearly without leaving any space for different interpretations. Also, the question is how robust the conclusions are when based on specific conditions that do not necessary represent the general aerosol types (see comments above on marine aerosol or coastal site case representativeness). On a positive note, the idea is very nice and useful, so improving methodology and providing concise recommendations/constraints in the discussion/separate section covering all aspects in one place will, in my opinion, deem this paper publishable in this journal.

Specific comments:

The conceptual diagram in Figure S1 might be better suited in the main manuscript, but some improvements are required.

Line 115: reference 16 is not appropriate for compressed film model as it uses an equilibrium gas–particle partitioning and liquid–liquid phase separation (LLPS) model with variable surface tension.

Figure 3: Y-axis name is missing in a) and b)

Lines 110-111: provide reference that Mace Head is representative of 'coastal marine with in-land air mass influence' site;

Line 216: acknowledge the source of MH 2012 data

Line 217: refer to specific chapters/figures in supplementary

Line 371: LLPS model was used in Ovadnevaite et al. not CF

References

- Davies, J. F., Zuend, A., and Wilson, K. R.: (2019) *Atmos. Chem. Phys.*, 19, 2933-2946.
- Facchini, M. C., Rinaldi, M., Decesari, S., Carbone, C., Finessi, E., Mircea, M., Fuzzi, S., Ceburnis, D., Flanagan, R., Nilsson, E. D., de Leeuw, G., Martino, M., Woeltjen, J., and O'Dowd, C. D.: (2008) *Geophys Res Lett*, 35, L17814.
- Gantt, B., Xu, J., Meskhidze, N., Zhang, Y., Nenes, A., Ghan, S. J., Liu, X., Easter, R., and Zaveri, R.: (2012) *Atmos Chem Phys*, 12, 6555-6563.
- IPCC: *Climate Change 2013: The Physical Science Basis. Contribution of Working Group I to the Fifth Assessment Report of the Intergovernmental Panel on Climate Change*, Cambridge University Press, Cambridge, United Kingdom and New York, NY, USA, 1535 pp., 2013.
- O'Dowd, C. D., Facchini, M. C., Cavalli, F., Ceburnis, D., Mircea, M., Decesari, S., Fuzzi, S., Yoon,

Y. J., and Putaud, J. P.: (2004) *Nature*, 431, 676-680.

Ovadnevaite, J., O'Dowd, C., Dall'Osto, M., Ceburnis, D., Worsnop, D. R., and Berresheim, H.: (2011) *Geophys Res Lett*, 38.

Ovadnevaite, J., Zuend, A., Laaksonen, A., Sanchez, K. J., Roberts, G., Ceburnis, D., Decesari, S., Rinaldi, M., Hodas, N., Facchini, M. C., Seinfeld, J. H., and C, O. D.: (2017) *Nature*, 546, 637-641.

Rinaldi, M., Fuzzi, S., Decesari, S., Marullo, S., Santoleri, R., Provenzale, A., von Hardenberg, J., Ceburnis, D., Vaishya, A., O'Dowd, C. D., and Facchini, M. C.: (2013) *J Geophys Res-Atmos*, 118, 4964-4973.

Ruehl, C. R., Davies, J. F., and Wilson, K. R.: (2016) *Science*, 351, 1447-1450.

Spracklen, D. V., Arnold, S. R., Sciare, J., Carslaw, K. S., and Pio, C.: (2008) *Geophys Res Lett*, 35.

Zieger, P., Väisänen, O., Corbin, J. C., Partridge, D. G., Bastelberger, S., Mousavi-Fard, M., Rosati, B., Gysel, M., Krieger, U. K., Leck, C., Nenes, A., Riipinen, I., Virtanen, A., and Salter, M. E.: (2017) *Nature Communications*, 8, 15883.

Preamble to all reviewers:

We would like to thank all reviewers for their constructive remarks and comments. We believe that especially the suggestions to better justify and revise the characteristic cases (MA, HYY and MH (now NE), Fig. 1 and 2 and Table 1) have helped improve our manuscript considerably. As such, in tandem with consideration of the salient point raised by reviewer 3 – that airmasses present at the Mace Head station are best considered maritime in origin – we have revised our representative cases in the following way:

Marine (MA):

1. **Aerosol size distribution** has been replaced with the annual average of the seasonal distributions reported by *Yoon et al. (2007)* from Mace Head.
2. **Organic mass fraction** (in both Aitken and accumulation modes) in the base case were maintained at 0.2, supported by three years of continuous submicron AMS measurements reported by *Yoon et al. (2007)*. Brief discussion on the variation in the chemical composition of marine aerosol has however been added with the appropriate references.
3. Surface-active **organic compound** replaced with palmitic acid (see e.g. *Tervahattu et al., 2002*; *Ngyuen et al., 2016* and references therein).

Boreal (HYY):

1. **Aerosol size distributions** were maintained the same as in the original manuscript, supported by e.g. *Häkkinen et al., 2012*.
2. **Organic mass fractions** were maintained the same as in the original manuscript, supported by e.g. *Häkkinen et al., 2012* and *Äijälä et al., 2019*.
3. The **organic compounds** in the accumulation and Aitken modes were represented by average properties of the SOA surrogate systems 1 and 2 given in Table S3.1 and S3.2 of *Ovadnevaite et al. (2017)*, respectively. These systems are representative of SOA from alpha-pinene ozonolysis and hence a reasonable proxy for boreal SOA composition (e.g. *Rastak et al., 2017*). Specifically, the dimers in the surrogate system 2 are thought to approximate the large molecules generated by the auto-oxidation mechanisms known to take place during alpha-pinene ozonolysis (e.g. *Ehn et al., 2014*).

Coastal (previously MH):

1. Inspired by the comments from Reviewer 3, we have removed this case and replaced it by the specific NUM-event (denoted NE) as studied and discussed by *Ovadnevaite et al.* Consequently, we have also moved Fig. 5 from the original manuscript to the SI, as we feel that it is not necessary for conveying the key messages of the study, while being potentially confusing regarding the marine vs. coastal nature of the Mace Head data.

The manuscript text and the Supporting Information (SI) have been revised accordingly. We would like to point out, however, that due to the extensive probing of the parameter space already done in the original manuscript; the implications of these changes for the conclusions of the study have been relatively minor. To further highlight the implications of the study for cloud properties and the indirect radiative effects of aerosol particles, we have incorporated Fig. S11 from the original supplementary into Fig. 4 (revised according to the descriptions above) of the revised manuscript, further complemented by a new Fig. 5 inspired by Reviewer 2's query regarding estimation of uncertainties in the smallest activated aerosol dry size. The new Fig. 5 addresses parametric uncertainties in several micro- and macro-physical cloud properties, as well as those in present in the surface phase induced shortwave cloud radiative effect (SW-CRE) differential, as modelled by the parcel model. It clarifies the scale-dependence of uncertainties associated with the existing theoretical framework of the parameterisation of ACI within climate models. Of particular note is the difficulty associated with resolving the bulk Köhler (BK) and approximate compressed film (CF) models within the parametric uncertainties present in the parcel model framework, yet the potentially significant SW-CRE perturbation.

Furthermore, we have corrected some typos and made a few similar minor edits to the text.

Our point-by-point responses to the reviewer comments and the related revisions to the manuscript are below. The reviewer comments are in *italics* and our responses are in normal font.

Reviewer #1:

The paper uses bulk Kohler and compressed film models in combination with an adiabatic parcel model to investigate the sensitivity of CDNC to aerosol size distribution and particle composition – specifically mass fraction and composition of organics. As the authors state, the results provide a basis for improving climate model simulations of ACI through the incorporation of molecular-level phenomena. In addition, the results should help guide collection of the most relevant observational data for a given region/aerosol type. The emphasis on the need for measurements that help constrain the organic aerosol budget is not new but the paper provides a strong case for the application of these measurements in reducing uncertainties in simulations of ACI. My comments primarily concern assumptions made about the composition of the three different aerosol types. I would have liked more detail on what went into the assumptions. This and other specific concerns are listed below.

We thank the reviewer for the encouraging comments. We have now revised the representative cases as outlined above and added a more thorough discussion and justification for the choice of the input parameters, together with the relevant references. Please see below for the responses to the specific comments.

Lines 118 – 120: This appears to be the only place in the manuscript and supplementary material where sources of the assumed composition for each aerosol type are described. Please provide more details about the assumed composition as a function of particle size. Why are the Aitken and accumulation mode compositions different for MH but not HYY and MA as shown in Figure 1b? This seems to contradict the information given in Table 1 where the accumulation mode organic fraction for MA and HYY are listed as zero. What is the choice of organic component based on for each aerosol type? Is the surface tension for each of these organic components the same as indicated by an assumed uniform surface tension value of 40 (Table 1)?

These are good points and consequently we have reviewed and revised the representative cases presented. Most of the concerns raised have been answered in the general response to all reviewers (see the beginning of the document), but below are some specific responses to the questions and concerns raised by this reviewer.

For MA and HYY, the Aitken and accumulation modes were assumed to be of the same composition, the F_{org} being 0.2 and 0.67 for MA and HYY, respectively, based on a review of the relevant literature (e.g. Yoon et al., 2007; Ovadnevaite et al., 2014; Häkkinen et al., 2012; Äijälä et al., 2019). The reviewer is correct that these compositions were incorrectly documented in Table 1 of the original manuscript, and these typos have now been corrected in the revised manuscript. While there are of course indications of size dependent composition in the submicron range, particularly for the marine case (e.g. Cavalli et al., 2004; Quinn et al., 2015), quantitative constraints for the organic fraction were difficult to establish in a robust manner from the present literature. Instead, we choose to demonstrate the sensitivity to this assumption through the extensive probing of the parameter space (see in particular Fig. 3 in the revised manuscript), and discuss the wealth of the relevant literature. As the reviewer correctly points out, a mode-resolved composition was used in the MH (coastal case) in the original manuscript, following the approach by Ovadnevaite et al. We follow this same approach in the NUM case presented in the revised manuscript. We have clarified these choices in the revised manuscript.

The organic component for MA (palmitic acid in the revised manuscript) was chosen based on the known abundance of fatty acids in marine aerosol (e.g. Tervahattu et al., 2002). For HYY, the organic component was chosen based on the surrogate species representing SOA from alpha-pinene ozonolysis, which is known to be a reasonable approximation of SOA from boreal forests (see e.g. Rastak et al., 2017). The same surrogate species were chosen to represent the NUM case in the revised manuscript, following the approach by Ovadnevaite et al. (2017).

Indeed, we maintain a constant value of 40 mNm⁻¹ for the pure organic surface tensions. We deem this a reasonable approximation, given that the known surface tensions of palmitic acid and components present in the boreal forest SOA are typically of the order of 30-40 mNm⁻¹ (see e.g. Chumpitas et al., 1999; Topping et al., 2007; Riipinen et al., 2007) – although of course highly variable and quite uncertain. The choice of the representative surface tension value has also been clarified in the revised manuscript, and the importance of this input parameter (along with others) demonstrated through the ranking with the Sobol algorithm (see Figs. S5-7). Nevertheless, we have added further $\Delta_{\text{CDNC}}(\gamma_{\text{org}}, \delta_{\text{min}})$ surfaces to the SI of the revised manuscript (see Fig. S3) so this work can provide a quantitative estimate of CDNC sensitivity to the film model parameters.

Lines 134 to 137: It is acknowledged that these changes in the activated particle size are small. Can some uncertainty be assigned to the changes in r^ ?*

Within the scope of this study it is possible to provide an estimate of parametric uncertainties in any model output associated with uncertainties in model input parameters. As such, we taken a global random (latin hypercube) sampling of parameter combinations within the ranges given in Table S1 of the original manuscript. The resulting range of values in model outputs (maximum supersaturation, smallest activated dry size, CDNC, liquid water path, cloud optical thickness and albedo, and the shortwave cloud radiative effect (SW-CRE) differential) are given in mean-relative dispersion terms (Fig. 5) and absolute terms Fig. S13. From Fig. S13 one can see, perhaps unsurprisingly, that these small difference cannot be differentiated from one another within the derived uncertainties for the studied cases. Nevertheless, the resulting relative dispersion (Fig. 5) in the SW-CRE differential is substantial. This we believe highlights a broader issue in which process-developments may indeed be of import on the global scale. While their significance on the cloud-scale may seem minor, it may be only so in a relative sense to the uncertainty a variability associated with meteorological and aerosol parameters. This feature of the cloud response naturally complicates e.g. the detection of signals from any aerosol-related changes in real-life cloud data.

Lines 143 – 144: Please state explicitly why the change in droplet size and CDNC for the CF simulations are largest for MH. Presumably it has to do with composition since the organic mass fractions at HYY are larger.

As discussed above, we have now removed the MH case. The reasons for the differences in the original manuscript were indeed partly related to the chemical compounds chosen (which have also now been revised) and the shapes of the size distributions. In particular, for the NE, the small condensation sink associated with the accumulation mode facilitates the larger S_{\max} (Fig. 2a), thus activating smaller particles. Despite smaller changes in r^* for NE than in HYY simulations, CDNC (and therefore droplet sizes) sensitivity for NE is greater due to the large Aitken mode concentration.

Figure 1b: Different colors should be used so that it is easier to tell which compound is which. Also – change to “b) ASSUMED chemical composition....”

This is a good point and we have revised the manuscript accordingly.

Line 723: Shouldn't this be accumulation (N2)?

Indeed, thank you for pointing this out. We have revised the manuscript accordingly.

Lines 229 – 232: What are the upper thresholds in N2 and N1 (300 and 1190 cm⁻³) as representative of unpolluted air masses based on?

A first filter was applied $N_2 < 300 \text{ cm}^{-3}$, loosely based on natural variability in marine accumulation modes see e.g. Partridge *et al.* and Heintzenberger *et al.* following which the 95th percentile was taken on N_1 to obtain 1190 cm^{-3} . This has now been clarified in the SI of the revised manuscript.

Figure 3: I found this figure to be exceedingly complicated because of the amount of information it contains. I am not sure how to simplify it, though, given the length constraints of the journal.

In light of our case revisions, the information content for Fig. 3 has now been reduced following the removal of the coastal instance, which hopefully addresses this comment by the reviewer.

Reviewer #2:

Overall, this is an interesting study addressing an important topic in a new way, but more details are required before this can be published. Specific comments follow.

We thank the reviewer for acknowledging the value of our work. As discussed above, we have now provided more details on the approach, specifically the justification of the representative cases chosen. Detailed answers to the specific points raised are given below.

The authors need to provide more details regarding how their results depend on the chosen condensed film parameters. Why 0.2 nm? The studies cited show a diversity of behavior, depending on the specific organic considered. It doesn't seem that differences in CF film parameters are considered in the sensitivity analysis. Is there a reason why not?

A base case film thickness of 0.2 nm was chosen based on the range of 0.16 – 0.30 nm provided in Ovadnevaite et al. (2017). Whilst the minimum film thickness likely has many molecular dependencies, we have chosen to maintain the same value across our cases owing to the considerable uncertainty in its value for our cases – as also pointed out by the reviewer. The sensitivity to the film model parameters was and is, however, demonstrated through the Sobol analysis (see SI). We have furthermore provided additional $\Delta_{\text{CDNC}}(\gamma_{\text{org}}, \delta_{\text{min}})$ surfaces to the supplementary information (Fig. S3) of the revised manuscript so this work can provide a quantitative estimate of CDNC sensitivity to the film model parameters (see also the response to reviewer 1 regarding the choice of the organic component surface tension).

More information is required regarding how the particle composition was assumed to vary with size. It is certain that the composition of the Aitken mode particles differs from the accumulation. Only for the coastal case do the authors seem to account for this. Also, they assume primarily NaCl for the marine case. But the Aitken mode is unlikely to be NaCl, instead more likely to be sulfate (perhaps even quite acidic, as in Ovadnevaite). Does this matter? And why is HYY succinic acid while MH is an ester dimer? (Note: the citation of Ovadnevaite is insufficient, as they didn't actually detect these molecules. This was a guessed at part of the interpretation, as best I can tell.) Also, why are they using Fulvic acid for the MA particles, but citing a paper that does not use fulvic acid and, in fact, suggests that fatty acids are not especially good at impacting activation. None of this seem particularly justified, but perhaps more important it is not clear how it matters since it seems that the CF parameters used are simply constants with $\delta = 0.2$ nm. So, does the variability in the organic composition only impact the Raoult term for the BK approach? This lack of clarity regarding composition is my main concern.

This is a fair point, and we have now carefully revised the studied cases and justified our choices with the existing literature (see the beginning of this document). Below are our responses to the specific questions raised.

Indeed, while the Aitken mode is expected to differ from the accumulation mode in composition, robust and quantitative constraints for these different fractions (which also add one more degree of freedom to the overall approach) are not easy to establish from the present literature for MA and HYY. This owes primarily to the fact that experimental techniques capable of reliably probing the Aitken mode composition independently of the larger particles are rare, and therefore any sufficiently long time series of such observations are difficult to find. We have however replaced the inorganic fraction in the Aitken mode of the MA simulations with Ammonium sulphate, as opposed to sodium chloride.

As discussed above, we have now revised the organic fraction in the HYY and NUM cases to be represented with the surrogate SOA mixtures used by Ovadnevaite et al. (2017). As also pointed out by the reviewer, the choice of these surrogates is actually even better justified for HYY as compared with the NUM case, since the mixtures are thought to resemble alpha-pinene SOA (Rastak et al., 2017). However, we chose to use these same compositions for the NUM organic fraction for the sake of consistency with Ovadnevaite et al. (2017).

The choice of a fatty acid (palmitic acid in the revised manuscript) as a representative of the marine organic fraction was based on a review of literature on the topic, which does suggest palmitic acid to be among the key organic compounds present in the marine surface microlayer and organic sea spray aerosol (e.g. Tervahattu et al., 2002).

We do acknowledge that representing the complex organic fraction with single compounds is a coarse simplification, but also representative of the level of detail present in e.g. global climate models (e.g. Tsigaridis et al., 2014). In our relatively simple (yet rather comprehensive as compared with the parameterizations used in the global models) approach, the identities of the organic molecules indeed affect primarily the Raoult term in the base

cases. The sensitivity to the other input parameters such as the assumed pure component surface tension and CF parameters is, however, demonstrated in the Sobol ranking and sensitivity surface plots (Figs. S4-7 and S3). These points and the justification for the chosen compounds is highlighted in the revised manuscript.

Relating to the previous comment, it is not totally clear to me that the authors have really covered a sufficient parameter space to make concrete conclusions regarding sensitivity. They have most certainly done quite a bit and their results provide important guidance. But the lack of exploration involving composition (of both the organics and the inorganics) is a limitation that is not sufficiently addressed.

We hope that the revisions made to the representative cases and their description in the revised manuscript now address these issues raised by the reviewer. We do feel that our study does probe a significantly larger fraction of the relevant parameter space as compared to the present literature on the topic.

I strongly suggest that the authors update the abstract to be a bit more specific to the study. When they state things like “The ACI” this can mean many things.

The abstract has been considerably revised, thank you for the suggestion.

L116: Does the BK case really represent the “maximum reduction of the Raoult effect”? Max reduction in what? Won't the Raoult effect be maximized in the BK case?

Thank you for pointing this out – this should be “maximum reduction in the Raoult term (water activity)”, i.e. maximum affect the organic can have on the water activity since all organic is assumed completely soluble. This sentence has been reworded.

L134: Would be clearer to state “The resulting decrease in the size of the activated particles for the Aitken or accumulation mode particles at each site is”

This is not quite what r^* is, here we are referring to the smallest dry aerosol size class that is activated, as opposed to the mode size of the droplet spectrum, to which I think the reviewer refers to (see also discussion of Fig. 2). We have added “dry” to the text before “particle” to clarify the sentence in question.

L141: The details regarding how the Ovadnevaite case are unclear. The authors need to be much clearer about how this is done (size-dependent composition? Dry size distribution)?

This has been clarified throughout with respect to the SOA surrogate system proxy compounds discussed in the preamble above. In brief, the dry size distribution and mode-resolved relative organic to inorganic mass loadings are taken from the AMS measurements reported by Ovadnevaite during the NUM-event.

L151: How is “Significance” determined?

On the order of the aerosol indirect effect. We have clarified this in the text.

L173: I feel as if the discussion around this section could benefit from mention of Dusek et al. (Science, 2006), who conclude that “size matters more than chemistry.”

Indeed. Although Dusek et al. was indeed referred to at the end of the discussion section of the original manuscript, we have added a reference to this work here as well in the revised manuscript.

L248: Suggest deleting the word “convoluted.”

Deleted.

It is generally indicated that the authors are using the “compressed film” model. But they are actually using an approximation of the compressed film model (as stated in Ruehl et al.). This should be clarified.

This misunderstanding has been corrected throughout the revised manuscript.

Reviewer #3:

Aerosols and clouds cause the largest uncertainty to estimates and interpretations of the Earth's changing energy budget (IPCC, 2013). The complexity, however, associated with elucidating the specific organic matter effects is unrivalled compared to other pollutants e.g. inorganic compounds. Carbonaceous (organic matter and black carbon) are not even properly represented in the present-day models. Especially the organic effect on surface tension that has been observed to be important in cloud activation, but is not currently accounted in regional or global climate models. The use of static pure water surface tension, despite actual composition information, is still predominant in these models. What complicates matters even more is that there is no unifying theory for all types of situations, or, in other words, there is no one size fits all model and the classical κ theory with water surface tension is suitable for numerous of applications, but neglecting the effects of surface tension causes significant errors in others. Hence, it is important to define situations when accounting for surface tension reduction is crucial and when simple kappa theory is justifiable. An attempt to present the quantitative criteria for when surface effects should be accounted for in the global models, in this manuscript, is, thus, highly appreciated and timely. However, I see some significant drawbacks for case selection on which these criteria are based.

We thank the reviewer for appreciating the potential importance of our work. We have now addressed the concerns and provide detailed responses to the issues (and the corresponding revisions to the manuscript) below. Please see also the preamble text in the beginning of this document for a description of the general revisions made based on suggestions from all three reviewers.

The main issue is with a representativeness of selected reference cases/studies for conditions representing selected environments. Very specific cases constrained to one location (MH, HYY) or old paper for MA that does not cover other aspects of marine aerosol (specifically organic fraction in sea spray aerosol) are not really supporting the ambitious aim of this manuscript. As I noted above, the task is very timely and needed, but has to be based on more robust environmental description.

As discussed in the preamble text, we have now carefully revised the representative environmental cases.

Heintzenberg et al. paper is conceptually old and do not even include organic matter in the marine aerosol composition. Assumption that sea salt and nssSO₄ with MSA are the main components of the marine aerosol might have been the knowledge of the time, but this manuscript focuses on surface tension effects by organics, so it is essential to account for new knowledge on marine organics, both primary (sea spray) and secondary. Moreover, Heintzenberg et al. paper refers to marine environment as being strongly affected by anthropogenic influences, which again was more true at the time, but several recent sulphur isotope papers show that the anthropogenic contamination has significantly reduced over the oceans due to major reduction in sulfur pollution over the continents itself. Maybe, the NASA-sponsored Atmospheric Tomography Mission (ATom) campaigns and papers can provide better information on marine aerosol size distributions and compositions or use coastal stations data filtered for pristine marine conditions can help here.

Similarly, Mace Head data are referred to as coastal with terrestrial influence, which is correct in general, but the cases presented in this manuscript correspond to exclusively marine aerosol sampled at Mace Head. E.g. Line 60 and elsewhere, NUM event in Ovadnevaite et al. is referred to as coastal location with terrestrial influence, while authors of the original paper refer to pristine marine air masses (Black carbon mass concentrations <5 ng m⁻³ confirm measurements within a pristine marine air mass). While a general classification of Mace Head as coastal location with terrestrial influence is justifiable when criteria of marine air masses are not applied, the reference in this case is not appropriate without arguing why the intended meaning of the authors is not accepted here. Similarly in the conceptual diagram (Figure S1), Mace Head is presented as coastal site with terrestrial influence, but then NUM case from Ovadnevaite et al. is not appropriate for general example.

On the same note, Mace Head, as being coastal location, would certainly have an impact from sea spray, but this is not reflected in Figure 1 b) or anywhere else in the manuscript. Hence, the confusion again, does Mace Head case refer only to particle nucleation mode, as shown in Ovadnevaite et al., or was it meant to be a general representation of the coastal area. If the latter, sea spray particles should also be included, especially in the accumulation mode. Moreover, the selection of sodium chloride as a proxy for sea salt for marine case is not reflecting the current knowledge of the sea spray as it is known that solubility, hygroscopicity and, thus, cloud activation for NaCl differ from that of sea salt (Zieger et al., 2017). Finally, it is not clear whether O'Dowd et al. 2004 reference was used for coastal particle composition or marine as it shows much higher OM contributions

than it is currently presented in Figure 1 for marine case; More recent studies show high fraction of marine organics in sea spray (O'Dowd et al., 2004; Facchini et al., 2008; Ovadnevaite et al., 2011; Rinaldi et al., 2013) and its global importance (Spracklen et al., 2008; Gantt et al., 2012), what effect would that have on the conclusions here for MA case, especially in lines 145-146?

These are all fair points. We have now replaced the MA size distribution and composition representation with those representing Mace Head (Yoon et al. (2007), Ovadnevaite et al.(2014)). We have added some further discussion on the organic mass fraction present in the marine aerosol, in the simulation setup section, in relation to these references and Fig. 3, but maintain our $f_{org} = 0.2$ for the marine case as now based on submicron Mace Head aerosol and AMS measurements made there (Ovadnevaite et al.(2014)).

When it comes to the hygroscopic properties of sea salt, we are of course aware of the work by Zieger et al. (2017). While their results indeed indicate that sea salt is somewhat less hygroscopic than NaCl (about 10-15% smaller growth factor at 95% RH, depending on the aerosol generation mechanism), this uncertainty is relatively small as compared with the other uncertainties considered in this work – especially since the effect is dramatically reduced in CCN activation as compared with hygroscopic growth at sub-saturated conditions. The potential climate impact of this effect is therefore likely to manifest itself in the direct rather than indirect radiative effects of aerosols. We have, however, added a sentence acknowledging this effect and citing the work of Zieger et al. (2017).

Elaborate on the conclusion in lines 143-144 stating that 'The implications of surface activity for cloud microphysics are hence more significant for the coastal instance, despite the larger organic mass fraction at HYY';

This conclusion now relates to the NUM-event. The heightened cloud microphysical susceptibility, relative HYY, results from the smaller condensation sink (N_2) permitting activation of more Aitken mode particles due to a steeper size distribution gradient at r^* . Some additional text has been added to the indicated lines of the revised manuscript to clarify.

Another major problem is with differentiation and proper use of Compressed Film (CF) vs liquid-liquid phase separation (LLPS) models, here, they are combined into one and named as CF despite being principally different. Ovadnevaite et al. paper uses an equilibrium gas-particle partitioning and liquid-liquid phase separation (LLPS) model with variable surface tension as opposed to compressed film model and these are not interchangeable. So which inputs did you use – based on CF or LLPS? Figure 1 d) is better suited to illustrate an equilibrium gas-particle partitioning and liquid-liquid phase separation (LLPS) model rather than compressed film model as the latter refers to continuous droplet surface coverage by OM with only the film thickness varying with droplet growth (the compressed film model predicts that a particle will reach Sc when the surface film decreases in thickness to the point that individual molecules begin to separate; i.e., a 2D phase transition occurs, and the surface tension no longer varies with increasing D_{wet})(Ruehl et al., 2016)). Be specific what are you refereeing to when talking about compressed film model.

This is a good point and deserves clarification. We use a simplified CF framework accounting for fractional surface coverage, which is essentially similar to the “Model 3” of Ovadnevaite et al. (2017). Figure 1d is therefore an accurate representation of our scheme. We do not, however, consider the gas-particle partitioning in the calculations which we believe would not be appropriate in our single-compound representation of the organic molecules. We have clarified this in the revised manuscript (see also responses to reviewer #2).

Also, CF model does not properly represent small particle effect (Davies et al., 2019), was this accounted in making conclusions for NUM particles (lines 224-226)?

We agree that further work is necessary to elucidate the validity of any approaches relying on bulk thermodynamics when discussing nano-scale particles. Accounting for such effects is, however, beyond the scope of this work that seeks to build on and elaborate on the wider implications of the conclusions drawn by e.g. Ruehl et al. (2016) and Ovadnevaite et al. (2017). We have deliberately chosen an approach that is consistent with the latter study to ensure taking discussing the conclusions drawn about e.g. the NUM event to a wider context.

Do you state that compressed film model results in smaller droplet diameter here (Lines 138-140)? Explain how it compares with CF model in (Ruehl et al., 2016) and LLPS in (Ovadnevaite et al., 2017) where they show an increase in wet diameter?

Yes indeed, this is indeed a salient point for the wider literature and, in our view one of the key virtues of this work.

While it is indeed true that the critical diameter D_c of a given aerosol type and size is generally increased (and critical supersaturation decreased) when incorporating any condensation-sink-increasing process into Köhler theory (see Fig 1d, Ruehl et al. 2016), specific knowledge of D_c is of only interest in determining kinetically realised droplet concentrations relative to thermodynamically viable droplet concentrations (CCN). We would like to stress that D_c , as discussed in e.g. Ruehl et al., does not relate to cloud microphysical or optical properties in any meaningful sense. In lines 138-140 we refer to the cloud droplet diameters ($\gg D_c$) which are critical for determination of optical properties and hence the radiative effects of the clouds. With the addition of the surface phase there is an associated dynamic suppression of cloud supersaturation development (Fig. 2a) resulting in a reduced S_{\max} . This is an important process to acknowledge since it reduces the droplet growth rates of activated droplets yielding the reduced droplets sizes (Fig. 2c). This reduction in droplet size compounded with the increased CDNC results in the cloud brightening that we report in this study.

The authors would further like to emphasise the importance of this mechanism in aerosol-cloud interactions, specifically in relation to CCN measurements at fixed supersaturation. Such measurements cannot account for dynamic supersaturation suppression, which limits the scope of conclusions arrived at in such experiments. In addition the mechanism is a likely explanation of much of the discrepancy between our CDNC enhancement (145 %) in the NUM-event and the CCN enhancement (“ten-fold”) reported by Ovadnevaite et al. (2017). Since the nascent ultrafine mode has such a large concentration ($N_1 = 2000$) any slight change in r^* (due to dynamic supersaturation suppression, for example), will have a significant impact on CDNC/CCN predictions. We have clarified and emphasized these points in the revised manuscript.

Finally, it is claimed that robust and quantitative constraints for conditions wherein the surface phase plays a significant role in ACI are presented, but it is not entirely clear from the manuscript what those conditions are. While reader might be able to draw them from figures and supplementary tables presented, the ambitious aim requires listing them clearly without leaving any space for different interpretations. Also, the question is how robust the conclusions are when based on specific conditions that do not necessary represent the general aerosol types (see comments above on marine aerosol or coastal site case representativeness).

We have now added a new Fig. 5 as discussed above and summarized the constraints given in the new Tables 2 and 3 (Tables S2 and S3 of the original manuscript) in the main text of revised manuscript. We hope these additions, together with the revision of the environmental cases, help to address this important point raised by this reviewer.

On a positive note, the idea is very nice and useful, so improving methodology and providing concise recommendations/constraints in the discussion/separate section covering all aspects in one place will, in my opinion, deem this paper publishable in this journal.

Thank you for this positive concluding remark. We have now carefully addressed all the points raised and hope the reviewer finds them satisfactory.

Specific comments:

The conceptual diagram in Figure S1 might be better suited in the main manuscript, but some improvements are required.

We have now revised Fig. S1 by removing the coastal case, correcting a typo (“BC” -> “CF” in the box illustrating the activation schemes) and clarifying the nature of the simplified CF model we use.

Line 115: reference 16 is not appropriate for compressed film model as it uses an equilibrium gas-particle partitioning and liquid-liquid phase separation (LLPS) model with variable surface tension.

Addressed.

Figure 3: Y-axis name is missing in a) and b)

Addressed.

Lines 110-111: provide reference that Mace Head is representative of 'coastal marine with in-land air mass influence' site;

As discussed above, Mace Head is now used as the marine case (Yoon et al., Ovadnevaite et al. 2014).

Line 216: acknowledge the source of MH 2012 data

Addressed.

Line 217: refer to specific chapters/figures in supplementary

Addressed.

Line 371: LLPS model was used in Ovadnevaite et al. not CF

Addressed throughout.

References

- Äijälä, M., Daellenbach, K. R., Canonaco, F., Heikkinen, L., Junninen, H., Petäjä, T., Kulmala, M., Prévôt, A. S. H., and Ehn, M.: Constructing a data-driven receptor model for organic and inorganic aerosol – a synthesis analysis of eight mass spectrometric data sets from a boreal forest site, *Atmos. Chem. Phys.*, 19, 3645–3672, <https://doi.org/10.5194/acp-19-3645-2019>, 2019.
- Davies, J. F., Zuend, A., and Wilson, K. R.: (2019) *Atmos. Chem. Phys.*, 19, 2933–2946.
- Ehn, M., Thornton, J. A., Kleist, E., Sipila, M., Junninen, H., Pullinen, I., Springer, M., Rubach, F., Tillmann, R., Lee, B., LopezHilfiker, F., Andres, S., Acir, I.-H., Rissanen, M., Jokinen, T., Schobesberger, S., Kangasluoma, J., Kontkanen, J., Nieminen, T., Kurten, T., Nielsen, L. B., Jorgensen, S., Kjaergaard, H. G., Canagaratna, M., Maso, M. D., Berndt, T., Petaja, T., Wahner, A., Kerminen, V.-M., Kulmala, M., Worsnop, D. R., Wildt, J., and Mentel, T. F.: A large source of low-volatility secondary organic aerosol, *Nature*, 506, 476–479, 2014.
- Facchini, M. C., Rinaldi, M., Decesari, S., Carbone, C., Finessi, E., Mircea, M., Fuzzi, S., Ceburnis, D., Flanagan, R., Nilsson, E. D., de Leeuw, G., Martino, M., Woeltjen, J., and O'Dowd, C. D.: (2008) *Geophys Res Lett*, 35, L17814.
- Gantt, B., Xu, J., Meskhidze, N., Zhang, Y., Nenes, A., Ghan, S. J., Liu, X., Easter, R., and Zaveri, R.: (2012) *Atmos Chem Phys*, 12, 6555–6563.
- IPCC: Climate Change 2013: The Physical Science Basis. Contribution of Working Group I to the Fifth Assessment Report of the Intergovernmental Panel on Climate Change, Cambridge University Press, Cambridge, United Kingdom and New York, NY, USA, 1535 pp., 2013.
- Nguyen, Q. T., Kjær, K. H., Kling, K. I., Boesen, T., and Bilde, M.: Impact of fatty acid coating on the CCN activity of sea salt particles, *Tellus B: Chemical and Physical Meteorology*, 69, 1304064, <https://doi.org/10.1080/16000889.2017.1304064>, 2017
- O'Dowd, C. D., Facchini, M. C., Cavalli, F., Ceburnis, D., Mircea, M., Decesari, S., Fuzzi, S., Yoon, Y. J., and Putaud, J. P.: (2004) *Nature*, 431, 676–680.
- Ovadnevaite, J., O'Dowd, C., Dall'Osto, M., Ceburnis, D., Worsnop, D. R., and Berresheim, H.: (2011) *Geophys Res Lett*, 38.
- Ovadnevaite, J., Zuend, A., Laaksonen, A., Sanchez, K. J., Roberts, G., Ceburnis, D., Decesari, S., Rinaldi, M., Hodas, N., Facchini, M. C., Seinfeld, J. H., and C. O. D.: (2017) *Nature*, 546, 637–641.
- Ovadnevaite, J., et al. Submicron NE Atlantic marine aerosol chemical composition and abundance: Seasonal trends and air mass categorization, *J. Geophys. Res. Atmos.*, 119, 11850–11863, doi:10.1002/2013JD021330, (2014).
- Rastak, N., Pajunoja, A., Navarro, J. C. A., Ma, J., Song, M., Partridge, D. G., Kirkevåg, A., Leong, Y., Hu, W. W., Taylor, N. F., Lambe, A., Cerully, K., Bougiatioti, A., Liu, P., Krejci, R., Petaja, T., Percival, C., Davidovits, P., Worsnop, D. R., Ekman, A. M. L., Nenes, A., Martin, S., Jimenez, J. L., Collins, D. R., Topping, D. O., Bertram, A. K., Zuend, A., Virtanen, A., and Riipinen, I.: Microphysical explanation of the RH-dependent water affinity of biogenic organic aerosol and its importance for climate, *Geophys. Res. Lett.*, 44, 5167–5177, <https://doi.org/10.1002/2017gl073056>, 2017.
- Rinaldi, M., Fuzzi, S., Decesari, S., Marullo, S., Santoleri, R., Provenzale, A., von Hardenberg, J., Ceburnis, D., Vaishya, A., O'Dowd, C. D., and Facchini, M. C.: (2013) *J Geophys Res-Atmos*, 118, 4964–4973.
- Ruehl, C. R., Davies, J. F., and Wilson, K. R.: (2016) *Science*, 351, 1447–1450.
- Spracklen, D. V., Arnold, S. R., Sciare, J., Carslaw, K. S., and Pio, C.: (2008) *Geophys Res Lett*, 35.
- Zieger, P., Väisänen, O., Corbin, J. C., Partridge, D. G., Bastelberger, S., Mousavi-Fard, M., Rosati, B., Gysel, M., Krieger, U. K., Leck, C., Nenes, A., Riipinen, I., Virtanen, A., and Salter, M. E.: (2017) *Nature Communications*, 8, 15883.
- Tervahattu, H., Hartonen, K., Kerminen, VM, Kupiainen, K, Aarnio, P; Koskentalo, T, Tuck, AF and Vaida, V.: New evidence of an organic layer on marine aerosols:, *J. Geophys. Res.* 107(D7), 4053, doi:10.1029/2000JD000282, 2002. Yoon, Y. J., et al. (2007), Seasonal characteristics of the physicochemical

properties of North Atlantic marine atmospheric aerosols, *J. Geophys. Res.*, 112, D04206, doi:10.1029/2005JD007044.

Zieger, P., Väisänen, O., Corbin, J. C., Partridge, D. G., Bastelberger, S., Mousavi-Fard, M., Rosati, B., Gysel, M., Krieger, U. K., Leck, C., Nenes, A., Riipinen, I., Virtanen, A., and Salter, M. E. (2017).

Reviewers' comments:

Reviewer #1 (Remarks to the Author):

The authors have improved the manuscript significantly by better describing the aerosol properties assumed for the MA, HYY, and NUM cases. I recommend publication with no further changes required.

Reviewer #2 (Remarks to the Author):

I find that the authors have done a thorough job of revising in response to the comments from all reviewers. Many aspects are certainly clearer now. This work is undoubtedly of broad interest and addresses an important topic, making it appropriate for Nature Communications. However, upon rereading I find that there remain a few important issues for the authors to address before this work can be published.

L123: O'Dowd (Nature, 2004) show for one case the organic fraction for the smallest particle bin in excess of 85%, more than the 63% mentioned here. They also note the increasing contribution of organic material as size decreases. Compositional variability, such as that discussed in this work, is more important for CCN for small particles, which are less likely to activate. The AMS measurements, which the authors use, are biased towards the accumulation mode (which the authors recognize in their response). Indeed, the choice of 0.2 is a conservative value (as stated), with this in mind.

L135: It is not entirely clear how the SOA surrogate systems are representative of SOA from alpha-pinene ozonolysis. The cited paper (Ovadnevaite) is unable to assess the origin of the organics in their study. They may be monoterpenes, but they may not be.

L138: Autoxidation does not necessarily lead to dimer formation, as implied here. Autoxidation leads to the formation of highly oxygenated monomers. Dimers are a consequence of cross-reactions.

L181: It would be helpful for the reader if the authors were to elaborate on the reason for the increased water vapor condensation sink.

L187: It would be helpful to state "radius" rather than "size" for clarity.

Fig. 1: For the marine case, can the authors speculate why they find such substantial reduction in the critical supersaturation when palmitic acid is used as the coating yet the organic fraction is only 20%? Recent observations using related compounds (Nguyen, Tellus B, 2017; Forestieri, ACP, 2018) showed that fatty acids have little impact on activation, despite being surface active. Are the properties specified here consistent with the statement that palmitic acid is being used? Certainly the MW and density are used. But is the minimum thickness appropriate?

Fig. 1d: Should there be more than three curves? For example, for the MA case, don't there need to be two curves because the ammonium sulfate and NaCl cases will differ in terms of the relationship between mass fraction, volume fraction, and consequently thickness? The same question holds for Fig. 2. Do the differences between sub-cases (e.g. the two MA compositions behave identically?) not matter? I think they do, reflecting the text at Line 188, but the caption does not make this clear.

Fig. 1d: In revisiting this figure, I'm struck by how much larger the droplets are here, compared to in Ovadnevaite et al. (Nature, 2017). True, Ovadnevaite use 41 nm dry particles, compared to 50 nm here. But the surface tension starts to fall in Ovadnevaite at 200 nm wet diameter (100 nm radius), whereas here for what is a nominally similar case the surface tension does not start to change until 500 nm radius (1,000 nm diameter). This seems to be a much larger difference than

one might expect for a dry diameter difference between 41 nm and 50 nm. Can this be explained?

Fig. 2C: It might be helpful if Fig. 2C x-axis said "Cloud droplet radius" rather than "particle radius" to distinguish from the starting distributions in Fig. 1.

L258: The statement here that "The molecular properties of the organic species will naturally proliferate in importance with increasing organic mass," could certainly use some additional details. In what way? What parameters in the model would reflect such changes? The density? The minimum thickness? Similarly, the statement on L323 about the necessity of "capturing the chemical...processes" would be more readily understood if further discussion were provided about what chemical factors matter. Is it mainly the organic fraction (the parameter probed here)? Or do the properties of the organic matter matter too?

Reviewer #3 (Remarks to the Author):

Authors have appropriately addressed most of the comments, however, I still have few concerns left as listed below:

Lines 125-127: 'The remaining aerosol mass is taken to be ammonium sulphate and sodium chloride in the Aitken and accumulation modes respectively' ammonium sulphate is usually present in the accumulation mode as well, justify the separation and discuss the effect/no effect of sulphate replacement by sea salt in the accumulation mode;

Line 127: somewhere here, provide a short sentence on cloud effects of NaCl usage instead of sea salt as presented in the reply to reviewer comments (see below);

 Original comment: Moreover, the selection of sodium chloride as a proxy for sea salt for marine case is not reflecting the current knowledge of the sea spray as it is known that solubility, hygroscopicity and, thus, cloud activation for NaCl differ from that of sea salt (Zieger et al., 2017).

Autor reply: When it comes to the hygroscopic properties of sea salt, we are of course aware of the work by Zieger et al. (2017). While their results indeed indicate that sea salt is somewhat less hygroscopic than NaCl (about 10-15% smaller growth factor at 95% RH, depending on the aerosol generation mechanism), this uncertainty is relatively small as compared with the other uncertainties considered in this work – especially since the effect is dramatically reduced in CCN activation as compared with hygroscopic growth at sub-saturated conditions. The potential climate impact of this effect is therefore likely to manifest itself in the direct rather than indirect radiative effects of aerosols.

We have, however, added a sentence acknowledging this effect and citing the work of Zieger et al. (2017).

-BTW I could not find this in the text, nor reference to Zieger in the reference list.

Line 134 and elsewhere; add reference numbers to the surnames where there are several references with the same first author in the reference list.

It seems to me that MA cases are representative of all seasons and the average case, thus, represents the yearly average, while HYY cases are more biased towards the warm/high biological activity periods and, mostly, summer. Quantify the implications or, at least, discuss these differences and the resulting implications.

Fig. 2c name x-axis as droplet radius, not particles as the latter implies dry particle diameter.

And finally, it seems that there is a confusion between the wet (droplet) diameter and dry aerosol critical diameter. My comment below was referring to the wet, thus, cloud droplet diameter, not the D_c as in the authors reply. Therefore, a discussion is still missing on how this compares to Ruehl paper stating that D_{wet} (cloud droplet) diameter with surface tension effect is substantially larger than the one predicted by kappa Köhler, see figure 1C and D in Ruehl or the text from it below:

'The functional form of D_{wet} versus S deviates substantially from k-Köhler predictions, with a $D_{wet,c}$ that is $\sim 50\%$ larger than predicted by constant korg for a given S_c . For succinic acid-coated AS (Fig. 1D), $D_{wet,c}$ is observed to be $1.8 \mu\text{m}$, which is substantially larger than predicted ($D_{wet,c} = 1.4 \mu\text{m}$), assuming $k_{org} = 0.31$. A similar difference was observed for malonic acid (Fig. 1C). For SOA-coated AS in Fig. 2, $D_{wet,c} = 1.3$ to $1.4 \mu\text{m}$, which is much larger than the korg predictions of $D_{wet,c} = 0.9 \mu\text{m}$. (Ruehl et al., 2016)'

 Original comment: Do you state that compressed film model results in smaller droplet diameter here (Lines 138-140)? Explain how it compares with CF model in (Ruehl et al., 2016) and LLPS in (Ovadnevaite et al., 2017) where they show an increase in wet diameter?

Authors reply: Yes indeed, this is indeed a salient point for the wider literature and, in our view one of the key virtues of this work.

While it is indeed true that the critical diameter D_c of a given aerosol type and size is generally increased (and critical supersaturation decreased) when incorporating any condensation-sink-increasing process into Köhler theory (see Fig 1d, Ruehl et al. 2016), specific knowledge of D_c is of only interest in determining kinetically realised droplet concentrations relative to thermodynamically viable droplet concentrations (CCN). We would like to stress that D_c , as discussed in e.g. Ruehl et al., does not relate to cloud microphysical or optical properties in any meaningful sense. In lines 138-140 we refer to the cloud droplet diameters ($\gg D_c$) which are critical for determination of optical properties and hence the radiative effects of the clouds. With the addition of the surface phase there is an associated dynamic suppression of cloud supersaturation development (Fig. 2a) resulting in a reduced S_{max} . This is an important process to acknowledge since it reduces the droplet growth rates of activated droplets yielding the reduced droplets sizes (Fig. 2c). This reduction in droplet size compounded with the increased CDNC results in the cloud brightening that we report in this study. The authors would further like to emphasise the importance of this mechanism in aerosol-cloud interactions, specifically in relation to CCN measurements at fixed supersaturation. Such measurements cannot account for dynamic supersaturation suppression, which limits the scope of conclusions arrived at in such experiments. In addition the mechanism is a likely explanation of much of the discrepancy between our CDNC enhancement (145%) in the NUM-event and the CCN enhancement ("ten-fold") reported by Ovadnevaite et al. (2017). Since the nascent ultrafine mode has such a large concentration ($N_1 = 2000$) any slight change in r^* (due to dynamic supersaturation suppression, for example), will have a significant impact on CDNC/CCN predictions. We have clarified and emphasized these points in the revised manuscript.

Preamble to all reviewers:

We would like to thank the reviewers for their encouraging and helpful comments on the revised manuscript.

Our point-by-point responses to the reviewer comments and the related revisions to the manuscript are below. The reviewer comments are in *italics* and our responses are in normal font.

Reviewer #1 (Remarks to the Author):

The authors have improved the manuscript significantly by better describing the aerosol properties assumed for the MA, HYY, and NUM cases. I recommend publication with no further changes required.

Thank you.

Reviewer #2 (Remarks to the Author):

I find that the authors have done a thorough job of revising in response to the comments from all reviewers. Many aspects are certainly clearer now. This work is undoubtedly of broad interest and addresses an important topic, making it appropriate for Nature Communications. However, upon rereading I find that there remain a few important issues for the authors to address before this work can be published.

We appreciate the encouraging assessment of the revised manuscript, and have done our best to thoroughly address the remaining issues identified by this reviewer.

L123: O'Dowd (Nature, 2004) show for one case the organic fraction for the smallest particle bin in excess of 85%, more than the 63% mentioned here. They also note the increasing contribution of organic material as size decreases. Compositional variability, such as that discussed in this work, is more important for CCN for small particles, which are less likely to activate. The AMS measurements, which the authors use, are biased towards the accumulation mode (which the authors recognize in their response). Indeed, the choice of 0.2 is a conservative value (as stated), with this in mind.

When discussing the limitations of our representative cases (lines 150-157), we have added a sentence regarding the fact that the AMS data is generally biased towards representing the larger particles measured. Furthermore, also along the lines of the comments of Reviewer #3, we clarified the temporal representativeness of the AMS data sets used for the MA vs. HYY cases.

L135: It is not entirely clear how the SOA surrogate systems are representative of SOA from alpha-pinene ozonolysis. The cited paper (Ovadnevaite) is unable to assess the origin of the organics in their study. They may be monoterpenes, but they may not be.

The organic fraction of surrogate mixtures presented in Ovadnevaite et al. (and used here for consistency) are based on Master Chemical Mechanism –derived calculations followed by estimates of their gas-particle partitioning for alpha-pinene ozonolysis, and a simplification of such a system (see the Supporting Information of Ovadnevaite et al. for a detailed description). Indeed, however, the origin of the organics in Ovadnevaite is not entirely clear but the surrogate systems studied are used as a rough approximation. We have chosen an approach that is consistent with Ovadnevaite et al. to demonstrate the consequences of this novel study on a broader level. We feel however, that e.g. re-evaluating the origins of the organics in the NUM event or its representation are out of the scope of this work. In the revised manuscript, we have clarified the uncertainty in the composition of the NUM event (lines 148-149), and added some discussion encouraging further studies on the size-dependent chemical composition of organic aerosol (lines 354-356).

L138: Autoxidation does not necessarily lead to dimer formation, as implied here. Autoxidation leads to the formation of highly oxygenated monomers. Dimers are a consequence of cross-reactions.

True. We thank the reviewer for pointing out this inaccuracy and have corrected it in the revised manuscript to be consistent with what is written in Ovadnevaite et al. (2017).

L181: It would be helpful for the reader if the authors were to elaborate on the reason for the increased water vapor condensation sink.

The sentence (lines 185-189 in the revised manuscript) has been reformulated to make clear that the increased condensation sink is a consequence of the heightened CCN activity, which results in a larger number of (smaller) growing cloud droplets and hence a larger surface area for the water vapour to condense on.

L187: It would be helpful to state “radius” rather than “size” for clarity.

Amended.

Fig. 1: For the marine case, can the authors speculate why they find such substantial reduction in the critical supersaturation when palmitic acid is used as the coating yet the organic fraction is only 20%? Recent observations using related compounds (Nguyen, Tellus B, 2017; Forestieri, ACP, 2018) showed that fatty acids have little impact on activation, despite being surface active. Are the properties specified here consistent with the statement that palmitic acid is being used? Certainly the MW and density are used. But is the minimum thickness appropriate?

This is a very good point. Indeed, we have chosen a film thickness for the marine organic aerosol consistently with the other cases (and e.g. Ovadnevaite et al.) due to scarcity of accurate knowledge on the exact identities and film thicknesses of the relevant compounds. As also discussed by e.g. Forestieri et al. (2018), palmitic acid is indeed probably not a perfect representative of marine organic aerosol, especially in terms of the film thickness required to reproduce observations. We have therefore modified the description of the marine organic aerosol to point out that a surrogate with MW and density of palmitic acid but a film thickness representative of a more CCN active compound / mixture was used (lines 128-129 and 162-164). Furthermore, as discussed above, we have further highlighted the need for constraining the exact identities and properties of the compounds comprising organic aerosol, and the uncertainty in the surface properties, citing also Forestieri et al. (2018).

Fig. 1d: Should there be more than three curves? For example, for the MA case, don't there need to be two curves because the ammonium sulfate and NaCl cases will differ in terms of the relationship between mass fraction, volume fraction, and consequently thickness? The same question holds for Fig. 2. Do the differences between sub-cases (e.g. the two MA compositions behave identically?) not matter? I think they do, reflecting the text at Line 188, but the caption does not make this clear.

Indeed it is the case that there are different Köhler (Fig. 1c) and surface tension (Fig. 1d) curves for the Aitken and Accumulation mode compositions. However, the differences in these curves between each mode for a given environment are not that significant. We therefore have chosen to omit accumulation mode curves so as to not over-complicate the figure, and clarify in the caption that these curves correspond to Aitken mode particles only since they are generally of more relevance for the differences between the BK and CF cases.

Fig. 1d: In revisiting this figure, I'm struck by how much larger the droplets are here, compared to in Ovadnevaite et al. (Nature, 2017). True, Ovadnevaite use 41 nm dry particles, compared to 50 nm here. But the surface tension starts to fall in Ovadnevaite at 200 nm wet diameter (100 nm radius), whereas here for what is a nominally similar case the surface tension does not start to change until 500 nm radius (1,000 nm diameter). This seems to be a much larger difference than one might

expect for a dry diameter difference between 41 nm and 50 nm. Can this be explained?

There seems to be some confusion about radius vs. diameter. If radii (or equivalently diameters) are compared, our results approximately agree with those reported by Ovadnevaite et al. In Figure 1d (and c) the ST curves are given for a dry radius (as opposed to diameter in the case 41 nm case presented in Ovadnevaite et al.). Here shown are the Köhler and surface tensions curves for a NUM-event Aitken mode particle of 20.5 nm dry radius (41 nm dry diameter) for comparison with the Ovadnevaite et al. figure which approximately agrees in terms of wet diameter, equilibrium supersaturation and surface tension. The remaining discrepancies are likely a result of the differences between the approximate compressed film model employed here and the LLPS model used by Ovadnevaite et al., and also the use of aggregate properties as a proxy for the SOA mixtures used herein versus the explicit mixtures used by Ovadnevaite et al.

Fig. 2C: It might be helpful if Fig. 2C x-axis said “Cloud droplet radius” rather than “particle radius” to distinguish from the starting distributions in Fig. 1.

This is a good point. However, it is not necessarily true that all particles shown in 2c are cloud droplets under the definition of $r_{\text{wet}} > r_c$. Specifically, for the largest particles the Köhler curves become very flat with large r_c that may not become kinetically realised, though from a practical standpoint they are so dilute so as to be considered cloud droplets on account of their optical properties and concentrations. We have therefore amended to “hydrometeor radius”.

L258: The statement here that “The molecular properties of the organic species will naturally proliferate in importance with increasing organic mass,” could certainly use some additional details. In what way? What parameters in the model would reflect such changes? The density? The minimum thickness? Similarly, the statement on L323 about the necessity of “capturing the chemical...processes” would be more readily understood if further discussion were provided about what chemical factors matter. Is it mainly the organic fraction (the parameter probed here)? Or do the properties of the organic matter matter too?

In the revised manuscript we have drawn attention to specific parameters, the minimum film thickness and pure organic component of the surface tension in particular, Fig. S3 in which this proliferation can be seen and the results of the Sobol analysis of the boreal case relative to the marine case.

Reviewer #3:

Authors have appropriately addressed most of the comments, however, I still have few concerns left as listed below:

Thank you for the encouraging response to the changes we made in the previous round of revisions.

Lines 125-127: ‘The remaining aerosol mass is taken to be ammonium sulphate and sodium chloride in the Aitken and accumulation modes respectively’ ammonium sulphate is usually present in the accumulation mode as well, justify the separation and discuss the effect/no effect of sulphate replacement by sea salt in the accumulation mode; Line 127: somewhere here, provide a short sentence on cloud effects of NaCl usage instead of sea salt as presented in the reply to reviewer comments (see below);

Original comment: Moreover, the selection of sodium chloride as a proxy for sea salt for marine case is not reflecting the current knowledge of the sea spray as it is known that solubility, hygroscopicity and, thus, cloud activation for NaCl differ from that of sea salt (Zieger et al., 2017).

Autor reply: When it comes to the hygroscopic properties of sea salt, we are of course aware of the work by Zieger et al. (2017). While their results indeed indicate that sea salt is somewhat less hygroscopic than NaCl (about 10-15% smaller growth factor at 95% RH, depending on the aerosol generation mechanism), this uncertainty is relatively small as compared with the other uncertainties considered in this work – especially since the effect is dramatically reduced in CCN activation as compared with hygroscopic growth at sub-saturated conditions. The potential climate impact of this effect is therefore likely to manifest itself in the direct

rather than indirect radiative effects of aerosols.

We have, however, added a sentence acknowledging this effect and citing the work of Zieger et al. (2017).

-BTW I could not find this in the text, nor reference to Zieger in the reference list.

Our apologies for missing this in our revision of the MS, and thank you for catching it! This has now been added (lines 126-127). It is also worth noting however that in the climate model simulations performed by Zieger et al. global CDNC values were not significantly influenced by variations in the hygroscopicity parameter which further justifies our choice (supplementary figure S4 in Zieger et al.).

Line 134 and elsewhere; add reference numbers to the surnames where there are several references with the same first author in the reference list.

Amended.

It seems to me that MA cases are representative of all seasons and the average case, thus, represents the yearly average, while HYY cases are more biased towards the warm/high biological activity periods and, mostly, summer. Quantify the implications or, at least, discuss these differences and the resulting implications.

This is a good point. Indeed, unfortunately only one of the eight AMS data sets included in the Äijälä et al. (2019) study has been recorded during the winter. However, these being the most comprehensive collection of AMS data reported from Hyytiälä to date, we choose to stick to the available data and have just noted this in the text (as also pointed by the second reviewer) (see line 132 in the revised manuscript). We have also acknowledged the need for continuous observations of size-dependent UF aerosol composition as an important topic for future work.

Fig. 2c name x-axis as droplet radius, not particles as the latter implies dry particle diameter.

Good point. However, it is not necessarily true that all particles shown in 2c are cloud droplets under the definition of $r_{\text{wet}} > r_c$. Specifically, for the largest particles the Köhler curves become very flat with large r_c that may not become kinetically realised, though from a practical standpoint they are so dilute so as to be considered cloud droplets on account of their optical properties and concentrations. We have therefore amended to “hydrometeor radius”.

And finally, it seems that there is a confusion between the wet (droplet) diameter and dry aerosol critical diameter. My comment below was referring to the wet, thus, cloud droplet diameter, not the D_c as in the authors reply. Therefore, a discussion is still missing on how this compares to Ruehl paper stating that D_{wet} (cloud droplet) diameter with surface tension effect is substantially larger than the one predicted by kappa Köhler, see figure 1C and D in Ruehl or the text from it below:

‘The functional form of D_{wet} versus S deviates substantially from k -Köhler predictions, with a $D_{\text{wet},c}$ that is ~50% larger than predicted by constant k_{org} for a given S_c . For succinic acid-coated AS (Fig. 1D), $D_{\text{wet},c}$ is observed to be 1.8 μm , which is substantially larger than predicted ($D_{\text{wet},c} = 1.4 \mu\text{m}$), assuming $k_{\text{org}} = 0.31$. A similar difference was observed for malonic acid (Fig. 1C). For SOA-coated AS in Fig. 2, $D_{\text{wet},c} = 1.3$ to 1.4 μm , which is much larger than the k_{org} predictions of $D_{\text{wet},c} = 0.9 \mu\text{m}$. (Ruehl et al., 2016)’

Original comment: Do you state that compressed film model results in smaller droplet diameter here (Lines 138-140)? Explain how it compares with CF model in (Ruehl et al., 2016) and LLPS in (Ovadnevaite et al., 2017) where they show an increase in wet diameter?

Authors reply: Yes indeed, this is indeed a salient point for the wider literature and, in our view one of the key virtues of this work. While it is indeed true that the critical diameter D_c of a given aerosol type and size is generally increased (and critical supersaturation decreased) when incorporating any condensation-sink-increasing process into Köhler theory (see Fig 1d, Ruehl et al. 2016), specific knowledge of D_c is of only interest in determining kinetically realised droplet concentrations relative to thermodynamically viable droplet

concentrations (CCN). We would like to stress that D_c , as discussed in e.g. Ruehl et al., does not relate to cloud microphysical or optical properties in any meaningful sense. In lines 138-140 we refer to the cloud droplet diameters ($\gg D_c$) which are critical for determination of optical properties and hence the radiative effects of the clouds. With the addition of the surface phase there is an associated dynamic suppression of cloud supersaturation development (Fig. 2a) resulting in a reduced S_{max} . This is an important process to acknowledge since it reduces the droplet growth rates of activated droplets yielding the reduced droplets sizes (Fig. 2c). This reduction in droplet size compounded with the increased CDNC results in the cloud brightening that we report in this study. The authors would further like to emphasise the importance of this mechanism in aerosol-cloud interactions, specifically in relation to CCN measurements at fixed supersaturation. Such measurements cannot account for dynamic supersaturation suppression, which limits the scope of conclusions arrived at in such experiments. In addition the mechanism is a likely explanation of much of the discrepancy between our CDNC enhancement (145 %) in the NUM-event and the CCN enhancement (“ten-fold”) reported by Ovadnevaite et al. (2017). Since the nascent ultrafine mode has such a large concentration ($N_1 = 2000$) any slight change in r^* (due to dynamic supersaturation suppression, for example), will have a significant impact on CDNC/CCN predictions. We have clarified and emphasized these points in the revised manuscript.

Our apologies for the confusion in relation to D_c and D_{wet} on the Köhler curves. Nevertheless the arguments presented in the previous author reply apply to both the D_c and wet diameters more generally. A direct comparison between figure 1 in Ruehl et al. and Figure 2c in the present work is not viable, as the wet diameter at a given point in time depends on the history of the parcel studied in terms of the supersaturation and the dynamics of the aerosol / cloud droplet population. The method used by Ruehl et al. exposes the growing droplets to supersaturations consistently larger than the critical diameter and aims to measure the equilibrium wet diameter for this supersaturation, hence likely exposing the population to a different supersaturation history as our parcel model. Furthermore, the composition and size distribution of the aerosol (and hence cloud droplet) populations studied here and by Ruehl et al. are different. These are in fact important reasons why a dynamic cloud model should be used to draw further conclusions about the resulting cloud droplet spectra rather than looking at populations exposed to constant saturation ratios. Specifically:

1. In the parcel model, the water vapor supersaturation is dynamically calculated and with all odds different from that of the thermal gradient chamber by Ruehl et al. that is determined by the conditions (temperature and water vapour source) within the chamber. Furthermore, for any simulation timestep in the parcel model, the BK and CF schemes expose hydrometeors to different ambient supersaturation. As such the hydrometeor growth rates differ between these schemes, and since in the CF scheme, as previously described, the ambient parcel supersaturation is suppressed relative to the BK scheme suggestive of reduced growth rates (though this influence reduced to some extent by the reduction in critical supersaturations in CF simulations); therefore the reduced droplet sizes in CF simulations relative to those in BK simulations are to be expected. The reduction in CF droplet sizes relative to BK increase in instances of more significant and persistent ambient supersaturation suppression (see NUM-event Fig. 2a and c). This highlights the need for caution when extrapolating results obtained from fixed supersaturation measurements into the context of dynamic cloud microphysics.
2. The organic fraction corresponding to the coating used chosen by Ruehl et al. is significantly larger (0.89) than used in our 3 cases.
3. Since the fraction of aerosol population that activate in under the CF scheme is generally increased, the condensed water vapour is distributed across a greater concentration of activated CCN. Therefore, one would expect smaller cloud droplets in the CF case even if the ambient supersaturation were the same in CF and BK cases.
4. The cloud droplet sizes shown in Fig. 2c are kinetically realised, as opposed to equilibrium sizes illustrated in Köhler curves.

REVIEWERS' COMMENTS:

Reviewer #2 (Remarks to the Author):

I have looked over the revisions and responses, and now find the paper suitable for publication.

Reviewer #3 (Remarks to the Author):

Authors have addressed all remaining concerns, therefore, I recommend the manuscript for publication. I would just ask to add a summary of discussion presented to the final question (on the critical diameters) to the manuscript or supplementary in addition to the reply.

Preamble to all reviewers:

We would like to thank the reviewers for their encouraging and helpful comments on the revised manuscript.

Our point-by-point response to the reviewer comments and the related revision to the manuscript are below. The reviewer comments are in *italics* and our responses are in normal font.

Reviewer #2 (Remarks to the Author):

I have looked over the revisions and responses, and now find the paper suitable for publication.

Thank you.

Reviewer #3:

Authors have addressed all remaining concerns, therefore, I recommend the manuscript for publication. I would just ask to add a summary of discussion presented to the final question (on the critical diameters) to the manuscript or supplementary in addition to the reply.

Thank you. To address the Reviewer's request, we have elaborated further on the importance of dynamic simulation of the supersaturation for the final droplet size (lines 295-301 of the revised manuscript) to complement the brief discussion that we had been incorporated on this topic in the previous version of the manuscript (lines 292-295 of the revised manuscript).